# Analogue signaling of somatodendritic synaptic activity to axon enhances GABA release in young cerebellar molecular layer interneurons

**Federico Trigo[1]\*[†], Shin-ya Kawaguchi[2]\*[†]**

[1]Departamento de Neurofisiología Celular y Molecular, Instituto de Investigaciones Biológicas Clemente Estable, Montevideo, Uruguay; [2]Department of Biophysics, Graduate School of Science, Kyoto University Oiwake-cho, Kyoto, Japan

**Abstract** Axons are equipped with the digital signaling capacity by which they generate and faithfully propagate action potentials (APs), and also with the analogue signaling capacity by which subthreshold activity in dendrites and soma is transmitted down the axon. Despite intense work, the extent and physiological role for subthreshold synaptic activity reaching the presynaptic boutons has remained elusive because of the technical limitation to record from them. To address this issue, we made simultaneous patch-clamp recordings from the presynaptic varicosities of cerebellar GABAergic interneurons together with their parent soma or postsynaptic target cells in young rat slices and/or primary cultures. Our *tour-de-force* direct functional dissection indicates that the soma-todendritic spontaneous excitatory synaptic potentials are transmitted down the axon for significant distances, depolarizing presynaptic boutons. These analogously transmitted excitatory synaptic potentials augment presynaptic $Ca^{++}$ influx upon arrival of an immediately following AP through a mechanism that involves a voltage-dependent priming of the $Ca^{++}$ channels, leading to an increase in GABA release, without any modification in the presynaptic AP waveform or residual $Ca^{++}$. Our work highlights the role of the axon in synaptic integration.

**\*For correspondence:**
ftrigo@iibce.edu.uy (FT);
kawaguchi.shinya.7m@kyoto-u.ac.jp (S-yaK)

[†]These authors contributed equally to this work

**Competing interest:** The authors declare that no competing interests exist.

## Editor's evaluation

This paper shows compelling experimental evidence for a novel mechanism for analog transmission of synaptic information from dendrites to distal nerve terminals in CNS interneurons. Brief subthreshold depolarizations caused by small EPSPs in dendrites are capable of activating the voltage sensors of presynaptic Ca2+ channels in nerve terminals, thus leaving them in a transient "primed" state. A subsequent suprathreshold action potential can then open these Ca2+ channels more efficiently leading to transmitter release facilitation. The effect decays within a few milliseconds and it is not dependent on residual Ca2+ levels or changes in presynaptic action potential waveform.

## Introduction

Synaptic integration stands at the core of neuronal signaling. During synaptic integration, neuronal information provided by the presynaptic neurons is processed, leading to a new encoding of signaling that takes into account both the activity of the presynaptic neurons and the intrinsic properties of the integrating neuron. In the classical view of synaptic integration, the tasks of various compartments of the neurons are sharply defined: the somatodendritic compartment gathers information

from presynaptic neurons; the axon initial segment sets the threshold for action potential (AP) firing; and the axon transmits the new AP to presynaptic terminals. In recent years, however, several studies have uncovered substantial deviations from this simple picture (where the different tasks that a neuron performs are canonically distributed between the different neuronal compartments), and today it is clear that individual neurons do not necessarily behave as the 'platonic' or 'canonical' neuron described by Coombs, Eccles, and Fatt in the middle 1950s (*Coombs et al., 1955*; *Llinás, 1988*; *Bucher and Goaillard, 2011*; *Goaillard et al., 2019*).

This conceptual evolution was in part due to the description of a significant electrical coupling between somatic and axonal compartments in the subthreshold voltage range (termed 'analogue signaling'). Although analogue signaling has been described in a variety of different preparations in mammals (for an exhaustive, recent review, see *Zbili and Debanne, 2019*), the quantification of the coupling with direct, simultaneous electrophysiological recordings at soma and axon terminals has been scarce in the literature because of the difficulties in recording from small varicosities of an intact axon in the majority of experimental preparations. As a corollary, analogue signaling has usually been studied by evaluation from indirect measurements and/or by strong subthreshold stimulation (using long, depolarizing, or hyperpolarizing voltage changes), so that the incidence, extent and physiological role of analogue signaling for subthreshold spontaneous activity is only known in a handful of neuronal types (*Alle and Geiger, 2006*; *Shu et al., 2006*; *Thome et al., 2018*). Exceptionally elegant direct axonal patch-clamp recordings have shown, both in the hippocampal mossy fiber > CA3 synapse (*Alle and Geiger, 2006*) and in synapses between layer 5 pyramidal cells (*Shu et al., 2006*), that subthreshold spontaneous or evoked somatodendritic activity can reach the axon. Such axonal integration of analogue signals coming from the somatodendritic compartment, when coupled to the AP-dependent signal, has been shown to affect the AP-dependent release (*Debanne et al., 2011*), and the resulting mixed or hybrid signaling mode has been called 'analogue–digital' signaling mechanism. In cerebellar molecular layer interneurons (MLIs), on the other hand, previous data obtained by paired somatic recordings from pre- and postsynaptic neurons suggest that subthreshold coupling between the somatodendritic and axonal compartments is also substantial (*Pouzat and Marty, 1999*; *Christie et al., 2011*; *Bouhours et al., 2011*; *Rowan et al., 2016*; *Rowan and Christie, 2017*), even for spontaneous activity (*Trigo et al., 2010*; *de San Martin et al., 2015*). However, the extent of such analogue signaling and its impact on synaptic outputs remain unclear because of the lack of a direct analysis by simultaneous recordings from pre- and postsynaptic structures in MLI synapses.

Cerebellar MLIs allow to quantitatively study the extent and functional impact of analogue signaling because their presynaptic bouton can be directly recorded with the patch-clamp technique, as shown by *Southan et al., 2000*, *Southan and Robertson, 1998*, and *Southan and Robertson, 2000* in pioneering, *tour-de-force* experiments. In the present work, we performed for the first time simultaneous electrophysiological, whole-cell patch-clamp recordings from the soma and the presynaptic boutons of cerebellar MLIs in order to assess the coupling under regimes of physiological activity and understand its physiological role. We quantitatively show that in young MLIs (PN13–17) the analogue coupling of somatodendritic synaptic activity to axon is substantial. By performing paired recordings from the presynaptic bouton and its postsynaptic target, we further show that this spontaneous synaptic activity coupled with an AP can affect transmitter release through a mechanism that involves the activation of voltage-dependent $Ca^{++}$ channels ($Ca_v$), with no change in the presynaptic AP waveform or basal $Ca^{++}$. Considered collectively with previous work, our findings highlight the importance of subthreshold coupling between neuronal compartments and further suggest that the process of synaptic integration is far more complex than classically envisaged, and that this is due to rich, largely unexplored signaling capabilities of the neuronal axon.

## Results
### Simultaneous soma–presynaptic bouton recordings in individual interneurons

In order to assess directly the degree of coupling between the somatodendritic and the axonal compartments we performed paired whole-cell recordings from the soma and the intact presynaptic varicosities of cerebellar MLIs in an acute slice at near-physiological temperature (34°C). To do so we first patched the somatic compartment with an intracellular solution (IS) containing the fluorescent

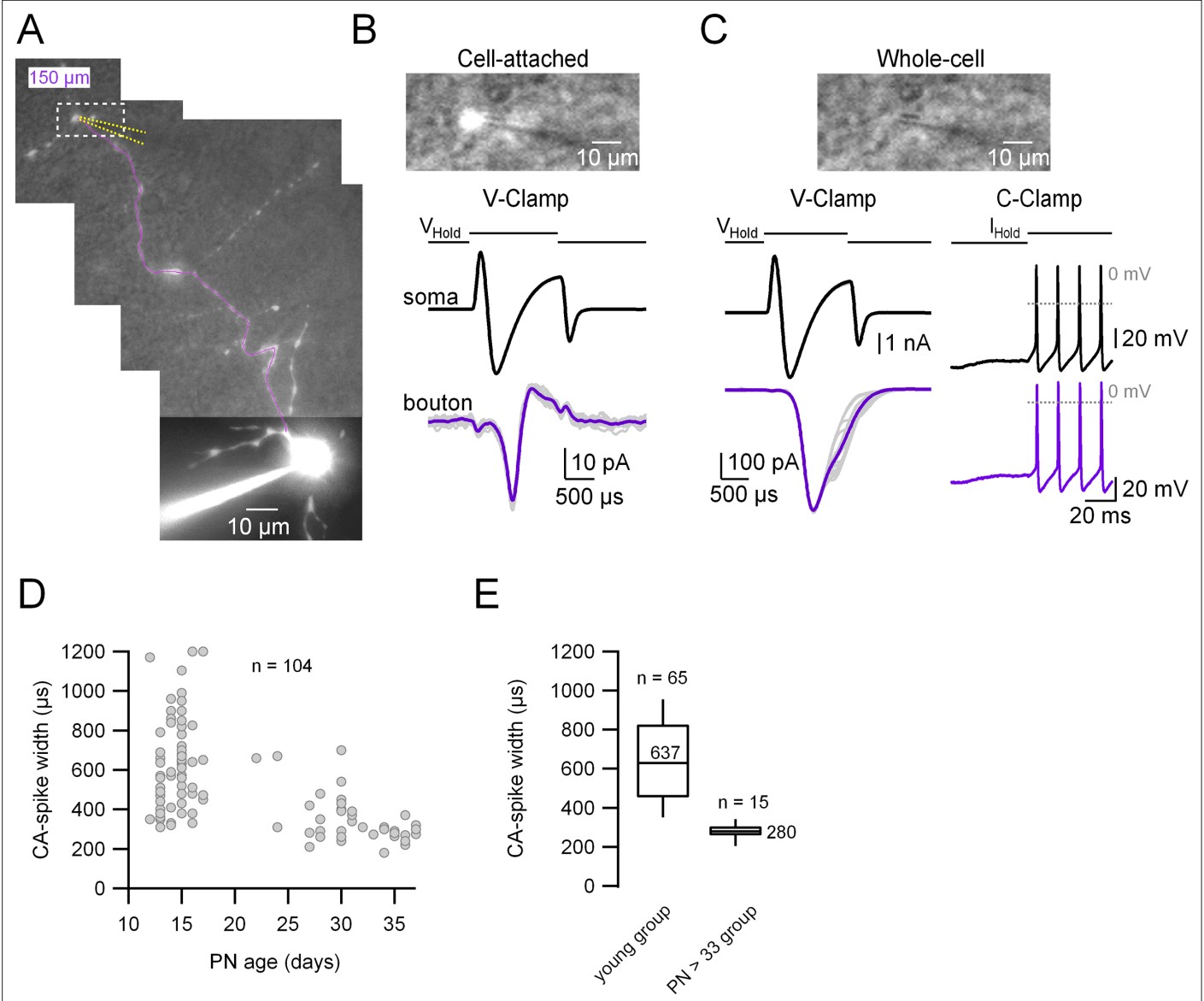

**Figure 1.** Simultaneous patch-clamp recordings from the soma and its corresponding axonal presynaptic terminal. (**A**) Fluorescence image of the whole-cell recorded molecular layer interneuron (MLI). The patched presynaptic varicosity is located at 150 µm from the soma. The position of the presynaptic bouton patch pipette is represented with dotted yellow lines and the main axon in magenta. Top: Simultaneous transmitted and fluorescent light pictures of the area shown with the dotted rectangle in (**A**), which shows the recorded presynaptic varicosity before (**B**) and after (**C**) rupturing the seal. In (**B**), the fluorescence image is saturated and does not reflect the real size of the presynaptic varicosity. Bottom: Somatic (upper) and presynaptic bouton (lower), voltage- (left) or current-clamp recordings (right) of somatically induced Na$^+$ current (or action potentials [APs]) and its propagation to the presynaptic varicosity. (**D**) Presynaptic AP width (measured from the cell-attached [CA] recordings) as a function of age (postnatal days 13–37). Total number of pairs = 104. (**E**) Box plots of the two age groups in which the recordings were categorized. Presynaptic CA-spike width of the immature age group (PN13–17): 639 ± 232 µs (median value 620 µs, *n* = 65). Presynaptic CA-spike width of the mature age group (PN33–37): 278 ± 44 µs (median value 280 µs, *n* = 15). Here, we set postnatal day 33 as the arbitrary limit between immature and mature animals.

The online version of this article includes the following source data for figure 1:

**Source data 1.** Data related to *Figure 1D and E* describing CA spike width.

dye Alexa 594; after waiting for a few minutes for the dye diffusion we went on to patch a presynaptic varicosity with a pipette containing the same IS without Alexa dye. In the example shown in *Figure 1A*, the distance between the center of the soma and the selected presynaptic varicosity is 150 µm (magenta line; minimum and maximum recording distances from the soma: 64.5 and 244 µm, respectively). The top panel in *Figure 1B* (corresponding to the dotted rectangle in A) shows the

recording configuration during the cell-attached mode, a few seconds before rupturing the patch, and the top panel in *Figure 1C* shows the same picture a few seconds after break-in, where the intra-varicosity fluorescence has disappeared. As shown in *Figure 1B, C,* the orthodromically transmitted presynaptic APs induced by a somatic depolarization were observed in cell-attached and whole-cell voltage and current-clamp (CC) configurations: in cell-attached (B); an unclamped (or escaping) spike current in voltage-clamp (VC; C, left), and a short train of APs recorded in CC (C, right).

Taking advantage of the cell-attached recording configuration presented above, which offers the possibility to characterize the presynaptic AP in unperturbed conditions (*Ritzau-Jost et al., 2021*), we attempted to determine whether the AP width changes with development (as has been shown for example in the Calyx of Held presynaptic terminal; *Taschenberger and von Gersdorff, 2000*). As can be seen in *Figure 1D, E*, the width of the presynaptic AP estimated from the width of spike current in cell-attached recordings (CA spikes) decreases with age (PN13–17: 639 ± 232 µs; PN33–37: 278 ± 44 µs). We set postnatal day 33 as the arbitrary limit between immature and mature animals because the presynaptic CA-spike width becomes less variable from that age onwards (CV of 0.3 for the younger age group; CV of 0.15 for the older group). Hereafter, all the experiments presented in the following sections were performed in the 'young' age group as defined from the analysis of the presynaptic AP width.

## Substantial somato-axonal coupling for synaptic activity

To quantify the coupling for spontaneous synaptic activity we performed two types of experiments. In the first, we monitored spontaneous excitatory synaptic potentials (sEPSPs) in paired somato-presynaptic bouton recordings. *Figure 2A* shows the recording configuration (top) and transmitted and fluorescent light pictures of the recorded presynaptic varicosity (bottom; which in this case was contacting a Purkinje cell soma). As can be seen in *Figure 2B, C*, the simultaneous CC recordings from the soma and the presynaptic varicosity show that every somatodendritic sEPSP is accompanied by the almost coincident appearance of an EPSP in the presynaptic varicosity (located here at 111.75 µm away from the soma). These presynaptic EPSPs are smaller in amplitude (mean ± standard deviation [SD] amplitude of somatic and presynaptic EPSPs in this cell: 13.1 ± 2.0 and 7.8 ± 1.6 mV, respectively; $n$ = 29 events, 50-s recording time, with an average coupling ratio [CR; see methods] of 0.6 ± 0.07), have a longer risetime (mean ± SD 10–90% risetimes of somatic and presynaptic EPSPs: 1.8 ± 0.49 and 5.7 ± 1.6 ms, respectively) and always appear later than the somatically recorded EPSPs (*Figure 2D*; lag in the cross-correlation between the somatic and presynaptic bouton recordings: 2.33 ms; inset in *Figure 2D*). All of these data indicate that the somatically recorded EPSPs are closer to the source of current, namely the dendritic postsynaptic densities, and are compatible with classical cable models for the propagation of subthreshold events in neuronal compartments. The analysis of the decaying phase of the presynaptic bouton and the somatic events shows that the axonal EPSP decay is well fitted with a mono-exponential function with a tau (20.4 ± 0.02 ms) which is extremely close to the slower time constant, $\tau 2$, of the somatic EPSP average (19.9 ± 0.02 ms). The fact that the final phase of decay, $\tau 2$, of the somatic and the presynaptic bouton events is the same, is also in accordance with the prediction by cable models: the decay cannot be slower than the membrane time constant of the cell (*Rall, 1969*).

In the second type of experiment, we performed spot laser photolysis of MNI-glutamate (*Canepari et al., 2001*; *Trigo et al., 2009*). Laser photolysis has various advantages in comparison to the recording of sEPSPs. First, the exact location of the activated dendritic varicosity can be measured, which allows a better quantification of the distance-dependent reduction of dendritic EPSPs while they propagate down the dendrites and the axon; second, a few repetitions can be made so as to perform averages and get clearer signals; finally, one can obtain responses in cells where the spontaneous EPSP frequency is low. *Figure 2E* shows the recording configuration: MNI-glutamate was photolysed in seven different dendritic varicosities plus the soma (*Figure 2E*; magenta spots). The laser-evoked EPSPs (eEPSPs) recorded simultaneously from the soma and the presynaptic bouton are shown in *Figure 2F*. As shown in *Figure 2G*, the soma–presynaptic bouton CR for eEPSPs at the different dendritic varicosities (and the soma) was not affected by the location of glutamate inputs at dendrites. The average CR calculated from the seven dendritic uncaging sites (dotted line in *Figure 2G*) was 0.4 ± 0.05 (mean ± SD; presynaptic bouton at 180.5 µm from the soma). As a comparison, the CR for a DC voltage stimulus (resulting from 400 ms current injections in the soma) in this cell was 0.71 (not

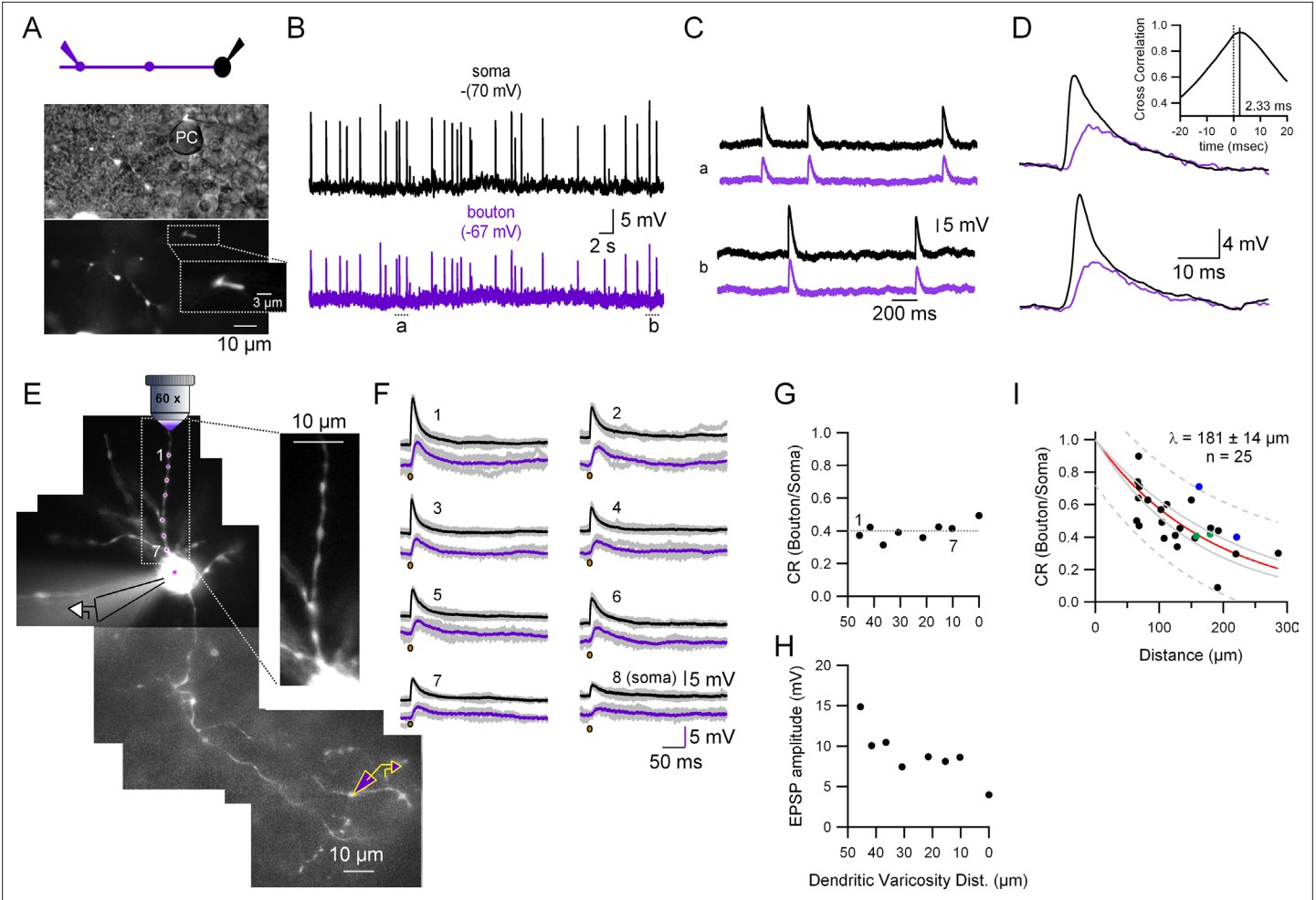

**Figure 2.** Quantification of the somato-axonal coupling for synaptic activity. (**A**) Top: Recording configuration. Bottom: Pictures highlighting the position of the recorded presynaptic varicosity on top of a Purkinje cell somata during the cell-attached configuration. The inset shows an expanded view of the varicosity showing the fluorescence inside the patch pipette. (**B**) Current-clamp (CC) recordings of spontaneous synaptic activity in the soma and the presynaptic bouton of the molecular layer interneuron (MLI). Resting membrane potential is indicated on top of each trace. (**C**) Two selected time epochs are shown (**a** and **b**) in (**B**) with an expanded time-scale. (**D**) Two selected pairs of subthreshold responses are shown with a ms time-range to see the details of the responses. The inset shows the cross-correlogram of the traces shown in (**B**) in the −20 to +20 ms interval. (**E**) Fluorescent picture of an Alexa 594-filled MLI showing the experimental configuration. Glutamate was photolysed from MNI-glutamate at different dendritic varicosities (magenta spots 1–7 plus soma) while simultaneously recording from the soma and presynaptic bouton. The inset, which corresponds to a single picture, shows that all the dendritic varicosities tested are at the same depth, which ensures the same laser intensity. (**F**) Somatic and presynaptic bouton CC recordings of the depolarizations evoked by the photolysis of glutamate at the different dendritic varicosities (and soma) shown in (**E**). Brown dots indicate the timing of the laser pulse. Laser pulse duration and power are 100 µs and 2 mW, respectively. (**G**) Coupling ratio (CR) between the soma and presynaptic varicosities for each tested dendritic varicosity as a function of distance of the stimulated site to the soma. Dotted line is the average. (**H**) Somatically recorded evoked excitatory synaptic potential (eEPSP) amplitude as a function of distance. (**I**) EPSPs amplitude ratios as a function of distance between somatic and presynaptic bouton recording sites. The continuous red line shows the fit with an exponential function. Data include experiments with spontaneous excitatory synaptic potentials (sEPSPs) and eEPSPs. Individual points correspond to the average ratio calculated from the average of all the detected presynaptic bouton and somatic EPSPs in each cell. Circles presented with the same color (blue and green) represent data from two different varicosities on the same axon. Dotted gray lines correspond to the 95% prediction bands and solid gray lines to the 95% confidence bands. The length constant ($\lambda$) value calculated was 181 ± 14 µm (mean ± standard deviation [SD]).

The online version of this article includes the following source data for figure 2:

**Source data 1.** Data related to *Figure 2* describing coupling between soma and axon.

shown). *Figure 2H* shows the amplitude of the somatically recorded dendritic EPSPs as a function of distance. It can be seen that the highest amplitude is obtained when glutamate is photolysed at the farthest dendritic varicosity (and the lowest amplitude when it is photolysed at the soma), indicating that the number of postsynaptic glutamate receptors is larger at distant dendritic varicosities than at

proximal varicosities, and that the amount of dendritic filtering in young MLIs is compensated by the receptor density gradient. In summary, glutamate photolysis experiments in MLI dendrites confirm that dendritic synaptic potentials with distinct amplitudes depending on the input location can travel down the dendrites to the soma with constant efficiency, and can further travel long distances down the axon to reach presynaptic terminals.

We next quantified the distance dependence of the somato-axonal EPSP coupling by measuring the EPSP CR as a function of distance in 25 different soma–presynaptic bouton pairs (*Figure 2I*; both spontaneous and laser-evoked EPSPs were used for the analysis). The length constant, $\lambda$, of this distance-dependent relationship is 181 ± 14 µm (with a 95% confidence interval of 29 µm). Considering that the total length of the MLI axon at PN14 is around 300 µm (*de San Martin et al., 2015*) and that the majority of the presynaptic varicosities is located in the proximal half (our unpublished observations), the data presented so far indicate that the analogue coupling between the somatodendritic and axonal compartments in young MLIs is prominent for synaptic activity.

## Impact of analogue–digital coupling on the synaptic output and AP waveform in MLI presynaptic boutons

It was shown before by our and other laboratories that analogue signaling for long (>hundreds of ms) depolarizing pulses can affect release. We thus wondered whether the analogically traveling spontaneous synaptic activity lasting only tens of ms (see *Figure 2*) could affect release as well. In order to assess whether this was the case, we performed paired whole-cell recordings between MLIs and the postsynaptic Purkinje cells. Alexa 594 fluorescent dye was included in the IS to allow for MLI visualization (*Figure 3A*). Once a connection was established, the MLI was held in VC and two types of interleaved protocols were applied: the control one consisted of a train of five stimuli (depolarization to 0 mV for 2 ms at 30 Hz) in order to induce somatic spikes; the test one was identical but the first AP-inducing pulse was preceded by a short (20 or 50 ms), subthreshold depolarization of the presynaptic MLI soma. From the postsynaptic current (PSC) train (*Figure 3B*), we calculated the amplitude ratio of the first response with vs without depolarization (*Figure 3C*) and the amplitude ratios of the subsequent PSCs in the train (*Figure 3D*). When the AP is preceded by a subthreshold depolarization the PSC amplitude is bigger than without the subthreshold depolarization (*Figure 3C*). Also, there is a decrease in the amplitude of subsequent PSCs (*Figure 3D*), which reflects an increase in the release probability at the first stimulation and a lower availability of release-competent vesicles at the subsequent stimuli. These results indicate that short, somatically applied subthreshold depolarizations right before the AP can increase release, as revealed by an increase in PSC amplitude and the resultant decrease of later release.

Previous studies in MLIs using longer (tens [*Rowan and Christie, 2017*], hundreds [*Christie et al., 2011*; *Rowan and Christie, 2017*], and thousands of ms [*Bouhours et al., 2011*]) depolarizations have proposed two primary mechanisms to explain the abovementioned phenomenon: a change in the presynaptic AP width or an increase in residual Ca++. By controlling the gating of presynaptic voltage-gated Ca++ channels (Ca$_v$), the duration of the presynaptic AP has a strong impact on the amount of released neurotransmitter (*Taschenberger and von Gersdorff, 2000*; *Geiger and Jonas, 2000*; *Sabatini and Regehr, 1997*; *Borst and Sakmann, 1999*; *Carta et al., 2014*; *Boudkkazi et al., 2011*), and modulation of the presynaptic AP duration in MLIs has been shown to happen as a result of analogue transmission from the somatodendritic compartments (*Rowan et al., 2016*; *Rowan and Christie, 2017*). In order to test whether these short, subthreshold depolarizations prior to an AP (pre-pulse) modulate the presynaptic APs, we performed cell-attached recordings of the presynaptic varicosities. We first studied the presynaptic AP under the same experimental paradigm of somatic stimulation presented in *Figure 3A–D*. After finishing the Purkinje cell recording, we attempted to record from a presynaptic MLI varicosity that was apparently contacting the postsynaptic Purkinje cell. For that aim, the postsynaptic Purkinje cell pipette was removed and replaced by a smaller-tip electrode in order to perform a cell-attached recording of APs from the presynaptic cell bouton. *Figure 3E* shows an example corresponding to the same MLI > Purkinje cell pair as in *Figure 3A*. After patching the MLI presynaptic bouton, the same (control and test) protocols were applied. Under these conditions, the width of presynaptic CA spike remained unchanged between the control and test protocols (*Figure 3F, G*; four recordings with prior postsynaptic Purkinje cell recordings and eight recordings where only soma–presynaptic boutons recordings were performed). The amplitude

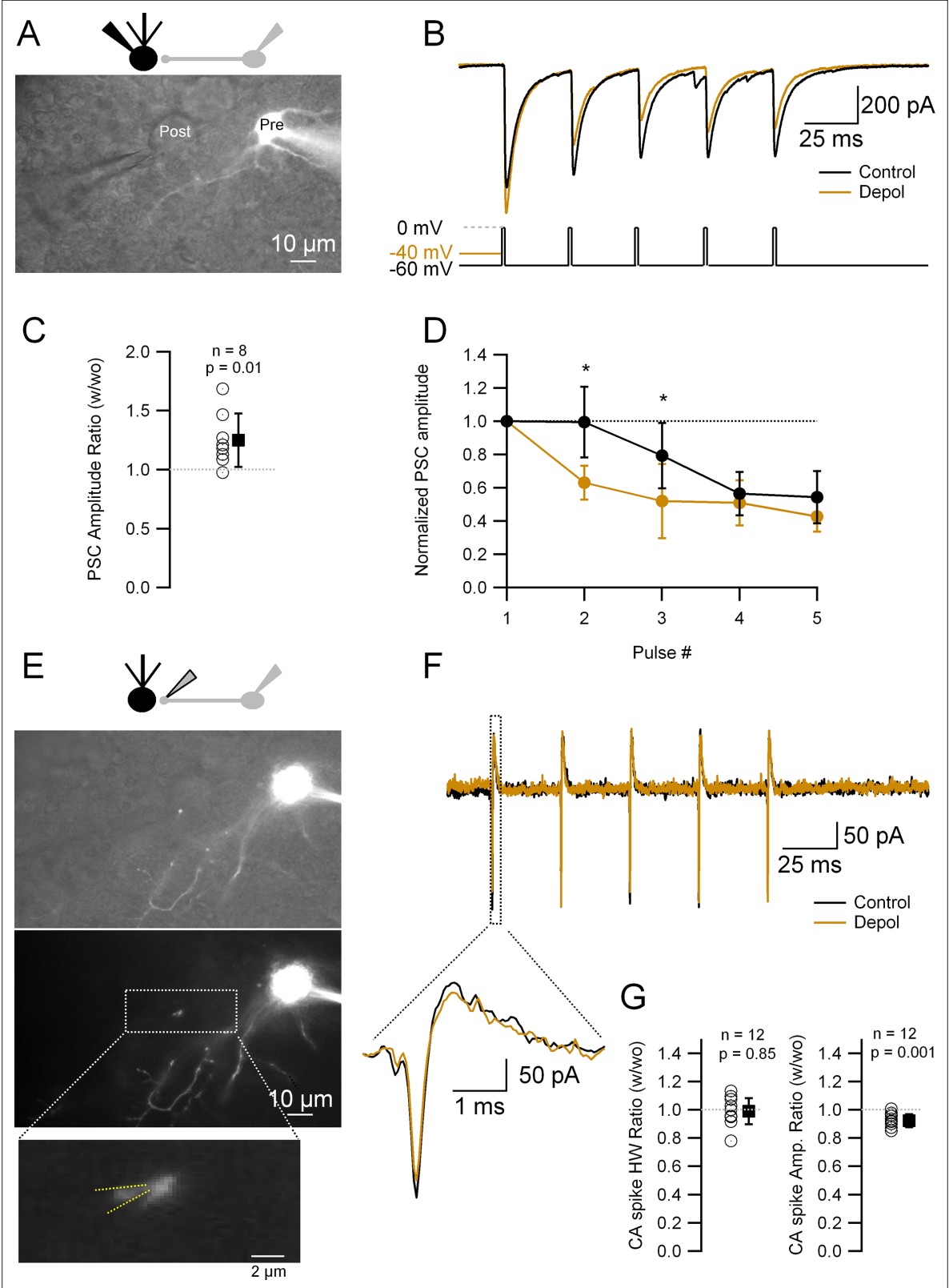

**Figure 3.** Short depolarizations increase postsynaptic current (PSC) and alter short-term plasticity but do not modify the presynaptic action potential (AP) width. (**A**) Top: Recording configuration. Bottom: Superimposed transmitted and fluorescence light pictures of the synaptically connected molecular layer interneuron (MLI) and postsynaptic Purkinje cell. (**B**) PSCs recorded from the Purkinje cell under two different experimental conditions: control (five APs evoked in the MLI by depolarization to 0 mV from a HP of −60 mV at 33 Hz; black trace); test (same but with a 20- or 50-ms depolarization to

*Figure 3 continued on next page*

*Figure 3 continued*

−40 mV before the first AP; brown trace). (**C**) Ratio of the first PSC in test over control conditions: 1.25 ± 0.23 (mean ± standard deviation [SD], *n* = 8, p = 0.01). Each empty circle corresponds to a single MLI > Purkinje cell pair. Black symbol corresponds to the mean ± SD values. (**D**) Amplitudes of PSCs in a train normalized over the first PSC in control and test conditions. Data show the averages ± SD from seven MLI > Purkinje cell pairs. Asterisks indicate statistical significance: p value for the first interval is 0.015 and for the second interval 0.016. (**E**) Top: Recording configuration. Middle: Superimposed transmitted and fluorescence light pictures highlighting the MLI presynaptic varicosity on top of the Purkinje cell soma. Same pair as in A. Bottom: Fluorescence light picture of the MLI. The inset shows the details of the recorded presynaptic varicosity. Yellow dotted lines show the approximate position of the presynaptic recording pipette. (**F**) Presynaptic APs recorded in the cell-attached configuration from the varicosity shown in (**E**) under the same experimental condition as in (**B**). The inset shows an expanded view of the first presynaptic APs in the train. (**G**) Ratio of the first CA (cell-attached) recorded spikes half-widths (HW) (left) and amplitudes (right) in test over control conditions: mean width ± SD = 0.99 ± 0.09 (*n* = 12, p = 0.85); amplitude mean ± SD = 0.92 ± 0.06 (*n* = 12, p = 0.001). Each empty circle corresponds to a single presynaptic varicosity. Black symbol corresponds to the mean ± SD values.

The online version of this article includes the following source data for figure 3:

**Source data 1.** Data related to *Figure 3* describing effects of pre-pulse depolarization on presynaptic APs and PSCs.

of the first presynaptic CA spike, on the other hand, showed a small, although significant, decrease (8%; *Figure 3F, G*), which is probably due to a decrease in the driving force for sodium influx during the AP onset because of the somatically elicited depolarization arriving at the presynaptic varicosity: given that the CR of DC signal is ~70% as noted above, the membrane potential ($V_m$) at the presynaptic varicosity would be 14 mV higher upon the 20 mV somatic depolarization, which is expected to decrease the driving force for sodium ions by about 10%. Altogether, these experiments indicate that the change in synaptic efficacy elicited by short, EPSP-like subthreshold depolarization prior to an AP does not involve any change in the presynaptic AP width.

To further examine the stability of presynaptic AP in MLIs, we also explored the main features of the presynaptic AP under other paradigms of stimulation. One of the main conclusions of these experiments is that AP propagation from the soma to the axon is extremely reliable in MLIs; indeed, we never observed propagation failures even at the highest firing rates attained (500 Hz), independently of whether the recording was performed from presynaptic varicosities located in the primary axon or axonal collaterals. *Figure 4* shows a representative example of a paired 'soma-bouton' recording where the soma is recorded in CC and stimulated at increasing stimulation intensities, and the propagated AP recorded in the presynaptic varicosity in the cell-attached configuration (*Figure 4A, B*). *Figure 4B* shows the instantaneous somatic firing frequency as a function of stimulus intensity for the 40 different trials. *Figure 4C* shows the raster plots for both the somatic and presynaptic bouton CA spikes, which indicate that every somatic spike is accompanied by the corresponding presynaptic one, even at the higher firing frequencies (350 Hz in this example). *Figure 4D, E* shows the half-width and amplitude, respectively, of the presynaptic, extracellular CA spike plotted as a function of the cell's firing frequency. It can be seen that the presynaptic CA-spike half-width is extremely stable, even at the higher (>300 Hz) firing frequencies (D) while the presynaptic CA-spike amplitude shows a small, although significant, reduction with frequency (E). In order to further study the relationship between half-width and firing frequency, we designed a paired pulse experiment where the soma was stimulated twice at decreasing time intervals (*Figure 4F*). Even at the shortest intervals, the presynaptic CA-spike width (second over first AP) does not vary. *Figure 4G, H* shows the results from 11 different soma–presynaptic bouton pairs. The presynaptic CA-spike amplitude was very stable as well, although a small decrease can be seen at the highest firing frequencies (*Figure 4H*, inset). This is in contrast to the somatically recorded AP, which shows a dramatic decrease in the amplitude at high but also low firing frequencies (black traces in *Figure 4A, F, H*). These results suggest that the AP is fully regenerated downstream of the AIS and they also show that the maximal firing frequencies that can be attained by stimulating the soma (*Figure 4A*) are much lower than the maximal firing frequencies that the axon can produce.

## EPSPs prior to an AP increase presynaptic Ca$^{++}$ influx and GABA release

The results presented above (*Figures 3 and 4*) indicate that transmitter release from MLI terminals can be modulated by a subthreshold, EPSP-like potential changes in the absence of any change in the presynaptic AP width. To study how the synaptic outputs from MLI boutons are augmented without

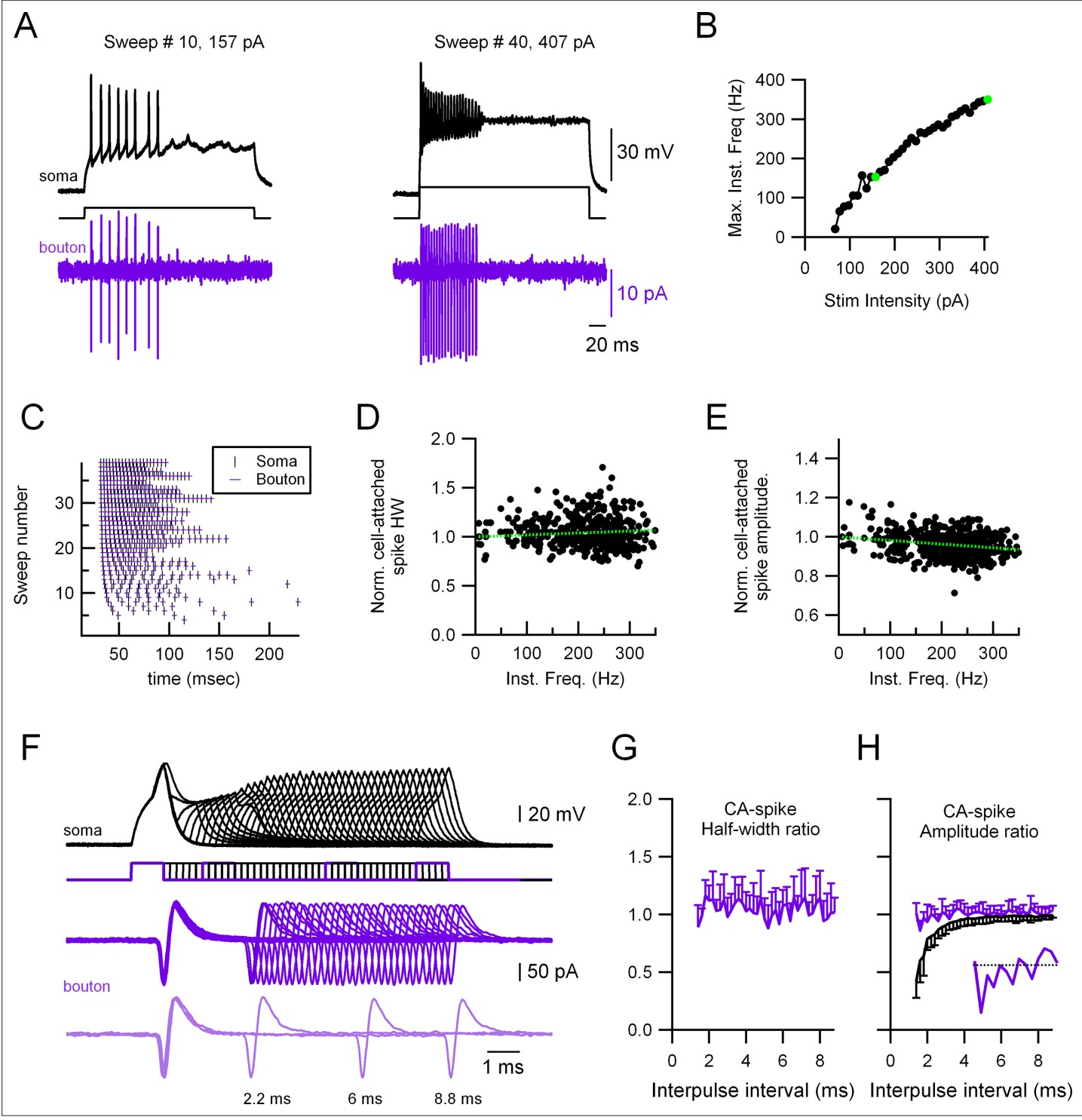

**Figure 4.** Highly reliable and stable presynaptic action potential (AP) in molecular layer interneuron (MLI) boutons. (**A**) Representative experiment showing a paired recording between the soma (whole-cell current-clamp; black traces) and the presynaptic varicosity (cell-attached in voltage-clamp; magenta traces). Firing was induced by injecting current through the somatic electrode (40 trials, 67–407 pA in 10 pA increments, 300 ms duration). Left: Responses at 157 pA stimulation intensity. Right: Responses at 407 pA stimulation intensity. (**B**) Maximal firing frequency measured from the somatic recordings. The green dots correspond to the sweeps shown in (**A**). (**C**) Raster plots of the somatic and the presynaptic bouton spikes for the 40 different trials, showing no presynaptic AP propagation failure. Normalized presynaptic CA-spike width (**D**) and amplitude (**E**) as a function of instantaneous frequency. Green, dotted lines show linear fits to the data. Spearman rank correlation test shows no significant correlation between presynaptic AP width and firing frequency (**D**) and a significant negative correlation (correlation coefficient = −0.27) between the presynaptic AP amplitude and firing

*Figure 4 continued on next page*

*Figure 4 continued*

frequency (**E**). The presynaptic AP widths and amplitudes are shown normalized to the first AP in each train to avoid errors due to fluctuations between trials. (**F**) Representative experiment showing a paired recording between the soma (whole-cell current-clamp; black traces) and the presynaptic bouton (cell-attached in voltage-clamp; magenta traces). Firing was induced by injecting twin current pulses at varying intervals through the somatic electrode (each current pulse was 1 nA and 1 ms duration). Light magenta, lower traces, show a selection of three trials at different time intervals (8.8, 6, and 2.2 ms). Ratios (second over first axonal AP) of CA-spike half-width (**G**) and amplitude (**H**) as a function of interpulse interval. Magenta traces correspond to the presynaptic CA spikes and black trace to the somatically recorded APs (in current-clamp). Data correspond to the averages of 11 different soma–presynaptic bouton pairs. Only either the positive or negative standard deviation (SD) is shown for clarity. The inset in (**H**) shows the presynaptic CA spikes amplitude ratio for the intervals 1.2–4 ms.

The online version of this article includes the following source data for figure 4:

**Source data 1.** Excel file for data related to *Figure 4* (panels B to E, G and H) describing presynaptic APs stability.

any change in the AP waveforms and given the technical difficulties associated with performing paired recordings from an MLI axonal bouton and its postsynaptic partner, we turned to the primary cerebellar culture preparation, where axon of interneurons and synaptic contacts can be sparsely eGFP (enhanced green fluorescent protein) labeled with an adeno-associated virus (AAV) vector. This allows to do simultaneous recordings of both a single MLI presynaptic varicosity and its postsynaptic cell and hence to perform a detailed analysis of the release at the synapse.

In order to test whether small depolarizations similar to spontaneous EPSPs can affect release we performed paired recordings of a single varicosity and its postsynaptic partner. The presynaptic varicosity was voltage-clamped and stimulated by a voltage waveform that consisted of either a single AP (*Figure 5A*, black, left traces) or an AP preceded by two consecutive EPSPs (*Figure 5A*, brown, right traces; for details see methods). Upon the application of a realistic stimulus represented by the control AP waveform, a presynaptic calcium current ($ICa^{++}_{pre}$) is induced that triggers a PSC (*Figure 5A*). When the AP is preceded by the EPSPs the corresponding $ICa^{++}_{pre}$ is larger and so is the PSC. *Figure 5B* depicts the temporal relationship of the onset (dotted line 'a') and the peak (dotted line 'b') of the $ICa^{++}_{pre}$ in relation to the AP, which shows that the difference between the $Ca^{++}$ currents in both conditions does not appear at the onset but at the peak of the current, during the decaying phase of the AP, suggesting that more $Ca_v$ are activated by an AP coupled with preceding EPSPs. *Figure 5C* shows the relative increases (AP waveform with over without EPSPs) of the $ICa^{++}_{pre}$ and PSC amplitude, that are both significant. During the course of our experiments we also patched a cerebellar granule cell terminal contacting an MLI. The same experiments performed in such a pair did not show any increase in the $ICa^{++}_{pre}$ nor PSC amplitude (*Figure 5—figure supplement 1A*).

To gain a better understanding of the relationship between the subthreshold potential and release, we performed a similar type of experiment to the one presented in *Figure 5A*, but now the $ICa^{++}_{pre}$ (and release) was triggered by a square depolarization (to 30 mV) preceded by 20 ms depolarizations to different voltages (−90 to −40 mV). *Figure 5D, E*, left, shows that the $ICa^{++}_{pre}$ (and the subsequent PSC) is highly dependent on the subthreshold voltage before the suprathreshold pulse: the more depolarized is the presynaptic $V_m$, the larger are the $ICa^{++}_{pre}$ and subsequent release. A complete statistical analysis of the responses induced by the different pre-pulses is shown in *Figure 5—figure supplement 2*, where the responses ($ICa^{++}_{pre}$ and PSC) obtained with the different pre-pulses were compared between each other. Again, there is no evidence of any activation of the $ICa^{++}_{pre}$ during the subthreshold depolarization. When a 3-ms interval is inserted between the subthreshold and suprathreshold pulses, the preceding subthreshold voltage levels do not have any influence on the $ICa^{++}_{pre}$ and transmission (*Figure 5D, E*, right). When the same protocol was applied to the granule cell terminal, there was no change in either of the two amplitudes ($ICa^{++}_{pre}$ or PSC; *Figure 5—figure supplement 1B*). In summary, taken together, these results strongly suggest that the subthreshold, short depolarization just before the AP impacts on the number of $Ca_v$ that are activated, leading to dynamic changes of the efficacy of synaptic transmission.

## Impact of subthreshold depolarization on $Ca^{++}$ channels in MLI boutons

To understand the mechanism by which the subthreshold depolarization increases the number of $Ca_v$ activated upon the arrival of a subsequent AP, independently from any change in the presynaptic AP width, we recorded the $ICa^{++}_{pre}$ in a bouton and performed a biophysical characterization of it (kinetics and activation voltage) upon depolarizing pulses. When the presynaptic bouton is voltage-clamped

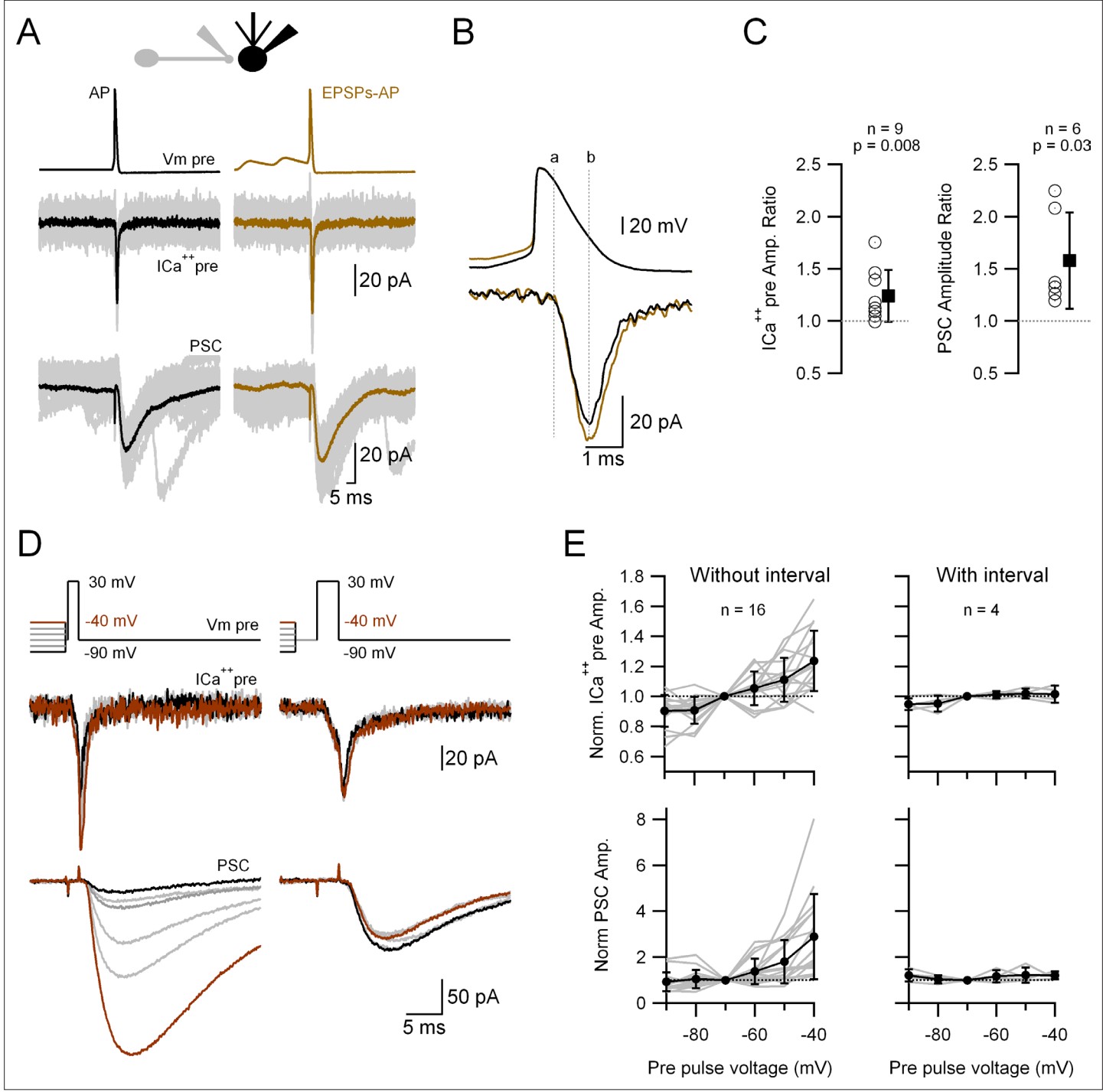

**Figure 5.** Passively propagated synaptic activity increases action potential (AP)-induced $ICa^{++}_{pre}$ and postsynaptic current (PSC) in cerebellar primary cultures. (**A**) Simultaneous, voltage-clamp (VC) recording of a presynaptic varicosity and the PSC. Upper traces show the voltage waveforms applied ($V_m$ pre), middle traces the recorded $ICa^{++}_{pre}$ and bottom traces the PSC. (**B**) Expanded AP waveforms (upper traces) and the resulting $ICa^{++}_{pre}$ (bottom traces). The 'a' and 'b' dotted lines correspond to the onset and peak of the $ICa^{++}_{pre}$, respectively. (**C**) Relative increases (waveform with EPSPs over that without EPSPs) of the $ICa^{++}_{pre}$ (left; 1.24 ± 0.25, $n = 9$, $p = 0.008$) and PSC (right; 1.58 ± 0.46, $n = 6$, $p = 0.03$) amplitudes. Each empty circle corresponds to a single varicosity ($ICa^{++}_{pre}$) or to a pair of varicosity > postsynaptic cell. Black symbol corresponds to the mean ± standard deviation (SD) values. (**D**) Simultaneous, VC recording of $Ca^{++}$ influx into a presynaptic varicosity (middle traces) and the PSCs in its postsynaptic partner (lower traces), without (left) or with (right) a 3-ms interval between the small and large presynaptic voltage pulses. The upper traces show the voltage waveforms applied. The suprathreshold depolarizations lasted 1.5 ms (left) and 3 ms (right). (**E**) Normalized $ICa^{++}_{pre}$ (upper graph) and PSC (lower graph) as a function of pre-pulse voltage for stimuli with (right) or without (left) a 3-ms interval between the pre-pulses and the suprathreshold depolarization. Gray lines correspond

*Figure 5 continued*

to individual varicosities and their corresponding postsynaptic cells. Black symbol corresponds to the mean ± SD values. For clarity, the complete statistical analysis of these results is presented in *Figure 5—figure supplement 2*.

The online version of this article includes the following source data and figure supplement(s) for figure 5:

**Source data 1.** Data related to *Figure 5A–C and E* describing augmentation of ICa++ and PSCs by subthreshould depolarization.

**Figure supplement 1.** Pre-pulse voltage does not influence release in the Granule cell axonal varicosities.

**Figure supplement 2.** Complete statistical analysis of the data presented in *Figure 5, D and E*.

---

to 0 mV for various durations, an inward $ICa^{++}_{pre}$ develops with an activation time constant of around 3.0 ms and little inactivation (*Figure 6A*; $n$ = 15). On the other hand, as shown in *Figure 6B*, the IV curve (from a total of 16 varicosities: 12 varicosities from primary cultures and 4 from slices; pooled because of similarity in both samples) indicates that the $ICa^{++}_{pre}$ starts to develop at Vm that are more depolarized than −40 mV, so it is unlikely that small depolarizations like the ones induced by EPSPs (as shown in *Figure 5*) would directly open the $Ca_v$, precluding the possibility of increased residual $Ca^{++}$ by the subthreshold depolarization as the mechanism for the analogue–digital coupling.

What is then the mechanism by which the analogue–digital coupling of EPSPs with an AP increases $Ca^{++}$ influx and release in MLI presynaptic boutons? To obtain an insight into the impact of subthreshold depolarization on $Ca_v$, we considered a biophysical model for $Ca_v$. The voltage-dependent $Ca_v$ open probability at steady state ($PCa_{v \ (stady \ op)}$) was simply expressed as a sigmoid function using two parameters, $V_{Cav50}$ (the potential for the half-maximal $Ca_v$ opening) and α (the factor reflecting steepness of the voltage-dependent activation), like Hodgkin–Huxley equations (for details of the model, see Materials and methods). Fitting of the $Ca^{++}$ current characterized by the *I–V* curve presented in *Figure 6B* with an assumption of $E_{Ca}^{++}$ value as +60 mV, yielded the two values for the free parameters: $V_{Cav50}$ = −17 mV and α = 0.2.

The α1 subunit of $Ca_v$ possesses four repeats of structural assembly, each of them having six transmembrane segments including the voltage-sensing S4 domains and the pore-forming regions of the channel. The (non-conductive) closed state of the channel can assume four different conformations, 'a' to 'd', represented in *Figure 6B*: the four voltage sensors are in their resting or non-activated position (a); there is 1 (b), 2 (c), or 3 (d) of the voltage sensors in their activated position. Here, we simply assumed that the four sensors are independently activated by voltage with an identical probability, $P_{s(act)}$ (but see *Hering et al., 2018*), and that the channel conducts only when the four voltage sensors are simultaneously active. The probability of the four sensors being in their active position simultaneously can be represented as the 4th power of $P_{s(act)}$. Thus, the probability of voltage-dependent steady activation of each voltage sensor, $P_{s(steady \ act)}$, can be represented as the 0.25th power of $PCav_{(steady \ op)}$. From the voltage-dependent probability change of $PCav_{(steady \ op)}$ (the red trace in *Figure 6B*), probability distributions of each state of $Ca_v$ are calculated as the curves shown in *Figure 6B*, bottom: $P_{s(steady \ act)}$ (the magenta trace), 'closed' (a, black trace), and 'primed' states ranging from (b) to (d) corresponding to closed $Ca_v$ with partially active states in some of the four voltage sensors. Evidently, these 'primed' states of $Ca_v$ clearly increase even at substantially hyperpolarized potentials.

The voltage-dependent state change of individual sensors between active and inactive states takes place as characterized by a certain velocity ($v$), which is dependent on the on ($k_{on}$) and off ($k_{off}$) rate constants (*equation (4)* in Materials and methods). Here, simulation of the $Ca_v$ opening and closure in the model to fit the time course of the actual presynaptic $ICa^{++}$ caused by a square depolarization pulse to 0 mV (*Figure 6A, C*) yielded the maximal rate constants $k_{on}$ and $k_{off}$ of 0.55 (/ms) and 0.65 (/ms), respectively. Consistently with previous studies showing that MLI boutons have axonal $GABA_A$ receptors causing autoreceptor currents upon presynaptic activation (*Pouzat and Marty, 1999*; *Trigo et al., 2010*; *de San Martin et al., 2015*), the presynaptic $ICa^{++}$ was accompanied with a slow kinetics, autoreceptor like current, which was isolated by the subtraction of the recorded $ICa^{++}$ and the simulated one (*Figure 6C*).

Using the biophysical $Ca_v$ model based on the above determined parameters, we studied whether the pre-pulse subthreshold depolarization indeed augments presynaptic $ICa^{++}$ as observed in the bouton recordings (as shown in *Figure 5*). Application of an AP waveform to the model resulted in an $ICa^{++}$ which was quite similar to the recorded $ICa^{++}$, and applying two consecutive subthreshold EPSPs just prior to the AP increased the simulated $ICa^{++}$ amplitude to 1.21-fold (*Figure 6D*), very

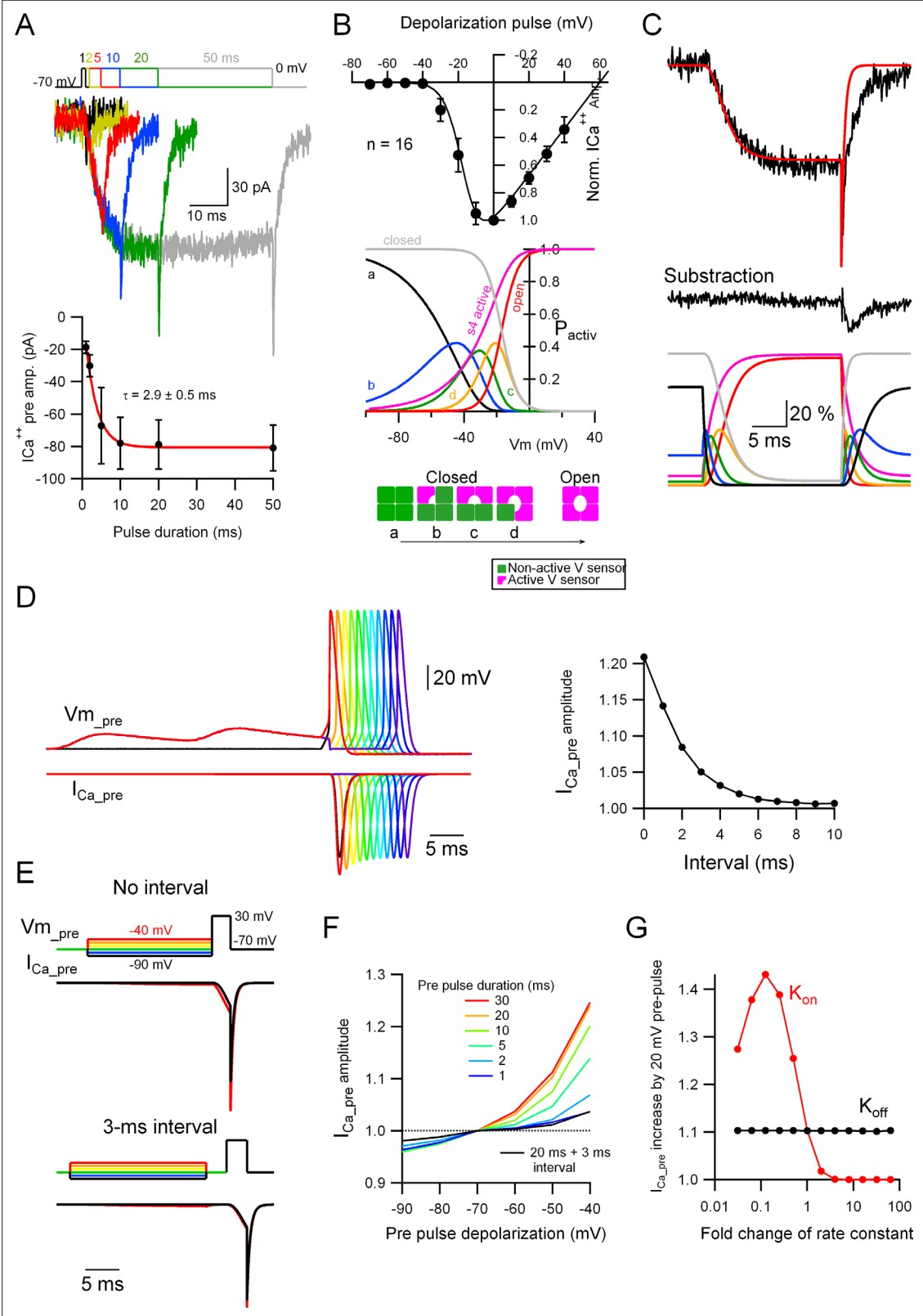

**Figure 6.** Subthreshold potential increases the probability of Ca_v voltage sensor activation. (**A**) Top: Voltage-clamp (VC) recordings showing representative traces of $ICa^{++}_{pre}$ upon depolarization to 0 mV for various durations (1, 2, 5, 10, 20, and 50 ms). Bottom: Average $ICa^{++}_{pre}$ amplitude as a function of stimulus duration (mean ± SEM). Red line is the fit to the data with an exponential function. Number of presynaptic boutons analyzed: 1 ms = 4; 2 ms = 8; 5 ms = 12; 10 ms = 16; 20 ms = 16; 50 ms = 16. (**B**) Top: The black circles show the current to voltage (*I/V*) relationship of the $ICa^{++}_{pre}$

*Figure 6 continued on next page*

*Figure 6 continued*

(mean ± standard deviation [SD]; *n* = 16) and the continuous line the simulated *I/V* relationship, which yielded the following parameters ($V_{Cav50}$ = −17 mV [the half-maximal Ca$_v$ voltage] and α = 0.2 [steepness of the voltage-dependent activation]). Middle: The probabilities of the different states of the Ca$_v$ channels are shown as a function of voltage. Bottom: Different states of the Ca$_v$ channels based on activation of independent four voltage sensors. (**C**) Top: Representative ICa$^{++}$$_{pre}$ (black trace) upon a depolarization to 0 mV for 20 ms and the simulated data (red trace). Middle: Subtracted data, which indicates that the difference between the experimental and simulated data probably corresponding to a GABA$_A$ autoreceptor current. Bottom: The probabilities of the different states of the Ca$_v$ channels upon the depolarizing pulse shown on the top, as a function of time. For the color code please see (**B**), middle and bottom schemes. (**D**) Left: Simulated data showing the presynaptic ICa$^{++}$ induced by a single presynaptic action potential (AP) and the effect of EPSPs right before the AP. Right: Effect of the interval between the prior EPSPs and the presynaptic AP on the ICa$^{++}$ pre augmentation. Please look at *Figure 5A–C* for a comparison with experimental data. (**E**) Simulated data showing the effect of a suprathreshold square depolarization on the presynaptic ICa$^{++}$ pre and the effect of pre-pulse, subthreshold depolarizations to various potentials. Please look at *Figure 5D, E* for a comparison with experimental data. (**F**) Effect of the duration and interval of pre-pulse depolarizations on the presynaptic ICa$^{++}$ pre augmentation. An interval between the subthreshold and suprathreshold pulses dramatically weakens the effect. (**G**) Effect of a change in the $k_{on}$ and $k_{off}$ rate constants on the presynaptic ICa$^{++}$ enhancement induced by a 20-mV subthreshold depolarization.

The online version of this article includes the following source data and figure supplement(s) for figure 6:

**Source data 1.** Presynaptic calcium current time constant vs recording series resistance, showing no correlation between both variables.

**Figure supplement 1.** Plots for ICa$^{++}$$_{pre}$ onset kinetics vs recording series resistance (Rs).

close to the actual experimental data (see *Figure 5C*). When an interval (ranging from 1 to 10 ms) was inserted between the EPSPs and the AP, the augmentation steeply disappeared, supporting the idea that the Ca$_v$ priming effect induced by the subthreshold depolarization rapidly occurs and vanishes (*Figure 6D*, right panel). Furthermore, the ICa$^{++}$ caused by a square pulse depolarization to 30 mV coupled with pre-pulse membrane potential changes of various sizes (as shown in *Figure 5D*) also resulted in the augmentation of the ICa$^{++}$, which was dependent on the size and duration of the pre-pulse (*Figure 6E, F*). When a 3-ms interval was inserted between the pre-pulse and the suprathreshold pulse the ICa$^{++}$ augmentation was almost completely abolished (*Figure 6E, F*), in line with the presynaptic bouton recording results (see *Figure 5D, E*). Surprisingly, the simulations indicated that the facilitating effect of the pre-pulse became evident even when its duration was as short as 5 ms (*Figure 6F*), suggesting that even brief depolarizations like a single EPSP would be sufficient to impact on the Ca$_v$ states. Finally, to examine the role of the kinetics of the channel sensor activation and de-activation on the ICa$^{++}$ augmentation, we systematically changed the parameters, $k_{on}$ and $k_{off}$, from 1/100- to 100-fold of the default value. As shown in *Figure 6G*, the activation kinetics $k_{on}$, but not $k_{off}$, has a clear impact on the ICa$^{++}$ augmentation, suggesting that the relatively slow activation of Ca$_v$ in MLI boutons is a critical factor for the analogue modulation of presynaptic ICa$^{++}$ and GABA release.

In summary, the experimental data and the kinetic simulation results indicate that priming of the Ca$_v$ by subthreshold potential changes in the presynaptic bouton is sufficient to explain the presynaptic ICa$^{++}$ augmentation and the resulting increase in GABA release. This mechanism is analogous to the one proposed previously by *Yang et al., 2014* for a K$^+$ conductance at the Calyx of Held synapse.

## Discussion

In this work, we performed simultaneous patch-clamp recordings from the soma and presynaptic varicosities of cerebellar MLIs. These experiments allowed us to measure directly the synaptic activity coupling between the two compartments. As anticipated from previous works in MLIs using other indirect methods (Ca$^{++}$ imaging; *Christie et al., 2011*; *Bouhours et al., 2011*), Ca$^{++}$ and GABA photolysis (*de San Martin et al., 2015*), voltage-sensitive dye imaging (*Rowan et al., 2016*), and electrophysiology (*Pouzat and Marty, 1999*; *Trigo et al., 2010*), we show here that the subthreshold voltage coupling in animals aged 12–17 days old is highly prevalent in these cells, even for short, subthreshold spontaneous activity (i.e., EPSPs). The somato-axonal analogue coupling of subthreshold physiological signals has been previously described by Alle and Geiger in the hippocampal dentate gyrus cells (*Alle and Geiger, 2006*), by *Shu et al., 2006* in L5 cortical neurons and by *Thome et al., 2018* in CA1 pyramidal cells. Interestingly, physiological signals can also travel antidromically, from the axonal boutons to the soma (*Paradiso and Wu, 2009*; *Trigo, 2019*). In this work, we further show that the EPSPs that reach the axon can modulate presynaptic Ca$^{++}$ influx and transmitter release through a

mechanism that relies on the priming of Ca$_v$ by voltage, independently from any change in basal Ca$^{++}$ or AP waveform.

## The mechanism by which orthodromic EPSP coupling affects release

It has been shown before by multiple groups, in cerebellar MLIs and in other neuronal types, that the somatodendritic membrane potential can be transmitted down the axon and affects transmitter release through the modulation of voltage-dependent conductances (*Zbili and Debanne, 2019*). Although a pure, AP-independent analogue type of transmission has been shown to occur in mammals and particularly in the retina (reviewed by *Heidelberger, 2007*), the subthreshold somatodendritic potentials transmitted to the axon can affect AP-dependent release and do not modify release per se. To stress the fact that the analogue and digital (AP) coding in axons act in combination, the term analogue–digital coupling has been coined in the literature. The analogue–digital coupling mechanisms may include a change in the AP waveform (amplitude and/or duration *Shu et al., 2006*; *Rowan et al., 2016*), a change of basal Ca$^{++}$ concentration (*Christie et al., 2011*; *Bouhours et al., 2011*) or other, non-defined mechanisms (*Alle and Geiger, 2006*; *Scott et al., 2008*). The term analogue–digital transmission is used to stress the fact that the subthreshold voltage (the analogue signal) before the arrival of the AP (the digital signal) can affect release. The results presented here show that: (1) the modulation of release does not require long or large voltage changes, but short and small, PSP-like voltage fluctuations of ≈10 mV can affect release; (2) the mechanism involved does not include any change in the presynaptic AP waveform or in the presynaptic Ca$^{++}$ influx right before the arrival of AP. Even in these conditions, the subthreshold depolarization before the presynaptic AP does induce an increase in the Ca$^{++}$ influx during the AP and hence an increase in release. Our experimental results, supported by a biophysical model of Ca$_v$ priming, are compatible with the following interpretation (schematized in *Figure 6*). During the subthreshold depolarization there is an increase in the voltage-dependent probability of the four voltage sensors of the Ca$_v$ to go from the 'resting-down' to the 'activated-up' position (*Hering et al., 2018*), but the pore of the channel remains closed. Once the AP is triggered, a bigger fraction of the voltage-gated Ca$^{++}$ channels opens in relation to the control (no depolarization prior to AP) condition where the probability of the voltage sensors to be in the activated position is at its minimum. The disappearance of the effect by a 3- to 5-ms interval after the subthreshold depolarization (*Figures 5D, E and 6D–F*) is compatible with this idea. In this sense, the mechanism described in this article is analogous to the K$^+$ channel facilitation described in the Calyx of Held terminal by *Yang et al., 2014*, who showed that synaptic activity can prime the K$^+$ channels through a mechanism that is exclusively dependent on the gating kinetics of the closed, intermediate states. Interestingly, the authors already predicted at the time that a similar mechanism may take place for other voltage-gated channels and in other synapses. Our work constitutes one such example. Although we cannot exclude that other mechanisms, as an increase in residual Ca$^{++}$ or a change in the presynaptic AP width, may operate in other circumstances leading to analogue–digital signaling in MLIs, they do not seem necessary to explain the effect described here. More importantly, these results highlight the relevance of recording directly from the axonal boutons in order to fully understand the mechanisms of synaptic release and AP propagation in CNS axons.

## Diverse analogue–digital coupling mechanisms in MLIs

Subthreshold coupling between the somatic and axonal compartments in MLIs was described in the 90 s by *Glitsch and Marty, 1999*, who showed that long somatic depolarizations can increase GABA release through a pure analogue transmission mechanism (similar to what happens in retinal amacrine cells). In 2011, *Christie et al., 2011* and *Bouhours et al., 2011* described for the first time an orthodromic subthreshold coupling in MLIs by which somatically induced depolarizations are transmitted down the axon and increase AP-evoked release through an augmentation in basal Ca$^{++}$, the so-called 'analogue–digital' transmission mechanism. It was later shown by *Rowan and Christie, 2017* that the analogue–digital increase in GABA release is dependent on a modulation of the presynaptic AP width because of the inactivation of a K$^+$ conductance (K$_v$3). Indeed, the presynaptic AP waveform is a critical determinant of synaptic strength because it sets the opening probability and duration of Ca$_v$ and hence of release probability. This has been shown in a variety of preparations like the giant brainstem auditory synapse, the Calyx of Held (*Borst and Sakmann, 1999*) (where both the pre- and postsynaptic compartments can be simultaneously voltage-clamped), the cerebellar granule cell

to Purkinje cell synapse (*Sabatini and Regehr, 1997*; *Kawaguchi and Sakaba, 2017*), Purkinje cell output synapses (*Kawaguchi and Sakaba, 2015*), and the hippocampal mossy fiber boutons (*Geiger and Jonas, 2000*; *Carta et al., 2014*; *Alle et al., 2011*). In the hippocampal mossy fiber boutons, different durations of the presynaptic AP activate the various subtypes of $Ca_v$ differentially according to their kinetics (*Li et al., 2007*). Apart from these effects on neurotransmitter release, the presynaptic AP waveform may also affect synaptic latency, as shown in cortical synapses between layer 5 neurons (*Boudkkazi et al., 2011*). In MLIs, direct electrophysiological and optical recordings of presynaptic AP have shown that voltage-gated $K^+$ conductances localized at the bouton can have a strong influence on the presynaptic AP waveform and release (*Rowan et al., 2016*; *Begum et al., 2016*).

Bouhours et al. showed, on the other hand, that the analogue–digital coupling was dependent on the activation of PKC (protein kinase C). In a recent work, (*Blanchard et al., 2020*) showed that the modulation of basal $Ca^{++}$ with local $Ca^{++}$ photolysis in a single MLI axonal varicosity can increase release through a modulation of the docking site occupancy. In the present work, both the amplitude and the half-width of the presynaptic AP are very resistant to various manipulations, like an increase in the stimulation frequency (*Figure 4*) and somatic depolarization (*Figure 3*), which is in agreement with what has been described by Alle and Geiger (see above) and by *Ritzau-Jost et al., 2021*, who showed that in neocortical (L5) fast-spiking GABAergic cells the width of the presynaptic AP does not change when the stimulation frequency increases. The fact that in our experiments the presynaptic AP width does not change, even when short pre-pulse depolarizations are applied before the AP (*Figure 3*), is in contrast to what has been found by *Rowan and Christie, 2017*, who showed that somatic presynaptic AP depolarizations slow down the presynaptic AP. The discrepancy may be due to a species difference (rat here vs mice in their work) or to the fact that the applied presynaptic depolarizations prior to AP are longer in their study than those applied in this study.

In MLIs, it seems plausible that multiple mechanisms by which somato-axonal coupling affects release coexist, each one of them probably being initiated under a particular condition (e.g., duration and amplitude of the somatic voltage changes). In this sense, it is interesting to note that the kinetics of the presynaptic $ICa^{++}$ is slow ($\tau$ around 3 ms, *Figure 6A*). As a comparison, the $ICa^{++}$ opens with an activation time constant of ~1 ms in Purkinje cell presynaptic boutons (*Kawaguchi and Sakaba, 2015*) and ~2 ms in cerebellar granule cells (*Kawaguchi and Sakaba, 2017*). This slow time constant of $Ca_v$ activation in MLI boutons, which is a critical factor for the 'priming' of $Ca_v$ as shown by our simulation (see *Figure 6G*), does not seem to be mainly due to poor VC conditions in presynaptic recordings. If this was the case, one would expect a correlation between the measured time constant and the Rs values: the higher the Rs, the slower the $\tau$. However, no correlation was found between the $\tau$ and Rs values measured from the 16 different varicosities presented in *Figure 6A* (see *Figure 6—figure supplement 1*, and Materials and methods). From the average Rs value calculated from those varicosities (82 M$\Omega$), on the other hand, an approximate VC time constant of 200 µs can be calculated, assuming a varicosity capacitance of 1–2 pF. As our simulation of $Ca_v$ demonstrated (see *Figure 6*), the $ICa^{++}$ recorded experimentally upon various distinct voltage waveforms, which should be differently affected by the clamp quality, could be nicely reproduced by the single parameter set for $Ca_v$ activation. Nevertheless, we should note that the slow kinetics of $ICa^{++}$ might partly arise from incomplete VC condition, particularly when presynaptic recordings are performed form a varicosity which is connected to neighboring varicosities with $Ca_v$ in a long axonal tract. In summary, the slow $\tau$ of the $ICa^{++}$ indicates that a very small fraction of the available $Ca^{++}$ channels opens upon the arrival of an AP in MLIs, leaving a big verge for different modulation mechanisms to occur, and the steep dependency of $Ca_v$ priming on the slow activation kinetics might alter the extent of $ICa^{++}$ augmentation by the pre-pulse voltage level depending on the exact channel subtype and various functional modulation of presynaptic $Ca_v$. Indeed, the large variability of $ICa^{++}$ pre augmentation by a pre-pulse (see *Figure 5C, E*) might reflect such a diversity of $Ca_v$ control among MLI boutons, a possibility which should be addressed in future studies. Unlike MLI boutons, a cerebellar granule cell bouton does not show an augmentation of $ICa^{++}_{pre}$ and transmission upon pre-pulse depolarization before AP (*Figure 5—figure supplement 1*).

### Ideas and speculation: the physiological role of somatodendritic spontaneous activity somato-axonal coupling

Studies of synaptic maturation in culture have shown that GABA signaling may participate in the regulation of synapse morphogenesis; reducing presynaptic GABA concentration, for example, leads to a deficit in GABAergic perisomatic innervation of cortical pyramidal cells (*Chattopadhyaya et al., 2007*). This suggests a positive retrograde action of released GABA on the maturation of presynaptic terminals (*Chattopadhyaya et al., 2007*; *Huang et al., 2007*). Our experiments have been performed at postnatal days 13–17, a time period where GABAergic synapses between MLIs and their postsynaptic partners (Purkinje cells and other MLIs) are still under maturation (*Ango et al., 2004*). Interestingly, during this time period spontaneous GABA release in MLIs gives rise to 'preminis', a form of autoreceptor currents that facilitates GABA release from MLI boutons and thus forms a positive feedback loop (*Trigo et al., 2010*; *Trigo et al., 2007*). Preminis are responsible for spike-evoked autoreceptor currents, and both autoreceptor currents and preminis are developmentally regulated, suggesting a specific role during maturation of GABAergic synapses (*Pouzat and Marty, 1999*; *Trigo et al., 2010*). In the light of these observations, it is tempting to speculate that both the somato-axonal coupling of spontaneous activity and the GABA$_A$ autoreceptors are part of the same activity-dependent mechanisms that link the production and release of GABA and the establishment of functional GABAergic contacts in the same neurons (*Ango et al., 2004*). In more general terms, analogue–digital coupling may be the common physiological behavior of short-axon neurons, at least at early stages of neuronal development.

## Materials and methods

**Key resources table**

| Reagent type (species) or resource | Designation | Source or reference | Identifiers | Additional information |
|---|---|---|---|---|
| Strain, strain background (*Rattus norvegicus, either sex*) | SD rat | Janvier Labs | RRID:RGD_38676310 | |
| Strain, strain background (*Rattus norvegicus, either sex*) | Wistar rat | IIBCE animal facility or Japan SLC, Inc | Slc:Wistar | |
| Transfected construct (adeno-associated virus) | Recombinant AAV2/9-eGFP | ***Kawaguchi and Sakaba, 2017*** | doi: 10.1016/j.celrep.2017.11.072. | AAV vector 2/9 to transfect eGFP in neurons |
| Chemical compound, drug | Tetrodotoxin Citrate | Tocris or WAKO chemical | Tocris: 1069/1 WAKO: 206-11071 | |
| Chemical compound, drug | Tetraethylammonium chloride | Tocris | Tocris: 3068/50 | |
| Chemical compound, drug | MNI-caged-glutamate | Tocris or HelloBio | Tocris: 1490/10 or HelloBio: HB0423 | |
| Chemical compound, drug | Alexa Fluor 594 Hydrazide | Thermo Fisher | Thermo Fisher: A10438 | |
| Software, algorithm | Fiji | Schindelin, J et al. https://doi.org/10.1038/nmeth.2019 | RRID:SCR_002285 | https://fiji.sc/ |
| Software, algorithm | Igor Pro | WaveMetrics | RRID:SCR_000325 | https://www.wavemetrics.com/ |
| Software, algorithm | Taro Tools | Labrigger, devloped by Dr. Taro Ishikawa | https://labrigger.com/blog/2011/07/21/taro-tools-and-ppt-for-igor-pro/ | https://sites.google.com/site/tarotoolsregister/ |
| Software, algorithm | Python Programming Language | https://www.python.org | RRID:SCR_008394 | |

### Preparation of cerebellar slices

Slices were prepared from Sprague-Dawley and Wistar rats of either sex aged 12–37 days old in strict accordance with the corresponding institutional guidelines (approval numbers A-750607 from Université de Paris and 001-01-2023 from CEUA [*Comisión de Ética en el Uso de Animales*], IIBCE). After

decapitation, the cerebellum was quickly removed in an ice-cold extracellular solution (ES), the cerebellum taken out and sagittal cerebellar slices (202 µm width) cut with a Leica vibroslicer (VT1200S). The slices were kept in a recovering chamber at 34°C until use. The composition of the ES was, in mM: NaCl 115, KCl 2.5, NaH$_2$PO$_4$ 1.3, NaHCO$_3$ 26, glucose 25, Na-pyruvate 5, CaCl$_2$ 2, MgCl$_2$ 1, pH 7.4 when bubbled with carbogen (95% O$_2$ and 5% CO$_2$). For animals aged 12–21 days the same aforementioned ES was used for the recordings. For animals aged above 21 days old a KGluconate-based ES was used for the dissection (**Blot and Barbour, 2014**); its composition (in mM): KGluconate 130, KCl 15, EGTA (ethylene glycol-bis(β-aminoethyl ether)-N,N,N',N'-tetraacetic acid) 0.05, HEPES (4-(2-hydroxyethyl)-1-piperazineethanesulfonic acid) 20, glucose 25, and D-AP5 50 µM, pH 7.4 and bubbled with carbogen.

## Preparation of cerebellar cultures

The method for preparing primary dissociated cultures of cerebellar neurons from wild-type newborn Wistar rats of either sex was similar to that in a previous study (**Kawaguchi and Hirano, 2007**), in accordance with the institutional guideline for animal experiments (approval number 202213 in Graduate School of Science, Kyoto University). Inhibitory interneurons were transfected with eGFP at 1 day after culture with AAV vector serotype 9 under the control of the CMV promoter (**Kawaguchi and Sakaba, 2017**). Interneurons could be visually identified from eGFP fluorescence. An axon of interneuron surrounding a PC soma was selected for whole-bouton recordings. Experiments were performed 3–5 weeks after preparation of the culture.

## Electrophysiology

Both the soma and axon of MLIs were recorded with the patch-clamp technique (**Hamill et al., 1981**) either in VC or in CC with a HEKA amplifier (EPC-10, double) and either Luigs & Neumann or Sutter manipulators. For the experiments presented in **Figures 1–4**, a KGluconate-based IS of the following composition (in mM) was used: 165 KGluconate, 10 HEPES, 1 EGTA, 0.1 CaCl$_2$, 4.6 MgCl$_2$, 4 Na$_2$ATP, 0.4 NaGTP, pH 7.3, and osmolarity 300 mOsm/kg H$_2$O. Alexa 594 (0.04 mM) was also added to the somatic recording pipette. With this IS the somatic pipettes had resistances of around 6 MΩ and the axonal ones 25 MΩ. To record the Ca$^{++}$ currents (**Figures 5 and 6**) a CsCl-based IS of the following composition (in mM) was used: 152 CsCl, 0.5 EGTA, 10 HEPES, 10.5 CsOH, 2 ATP, 0.2 GTP, pH 7.3, and osmolarity 300 mOsm/kg H$_2$O. With this IS the somatic pipettes had resistances of around 4 MΩ and the axonal ones 18 MΩ. In these experiments TEA (tetraethylammonium) (2 mM) and TTX (tetrodotoxin) (200 nM) were added to the ES in order to block K$^+$ and Na$^+$ voltage-dependent conductances.

The average series resistance (Rs) in presynaptic bouton recordings was 82 ± 31 MΩ (mean ± SD; $n$ = 16, calculated from the recordings shown in **Figure 6A**) and compensated by 30%. The recordings from presynaptic boutons were short lived (usually less than 5–6 min) and the Rs remained constant during this time. However, the experiments were discarded when the Rs value changed more than 20%. In order to address the possible VC issues associated with recording with a patch pipette from such a small structure, we performed an analysis of the relationship between the Rs value and ICa$^{++}$$_{pre}$ onset kinetics (the time constant of the rising phase). This analysis, presented in **Figure 6—figure supplement 1**, shows that there is no correlation between these two values. This, together with our experimental and modeling results, indicates that the quality of VC of the presynaptic boutons is not the cause of slow ICa$^{++}$$_{pre}$ kinetics.

Somatic and axonal pipettes were pulled with either HEKA (Pip 6) or Narishige (PP-83) vertical pullers. For experiments in the slice, we first recorded from the soma with the Alexa 594 containing IS. After a 5- to 10-min waiting time the fluorescence illumination was turned on in order to identify the axon and a suitable, superficial varicosity. In order to patch the axon a second pipette containing the IS without Alexa was used. The axon was patched by looking at the image created by the camera with both the bright field and fluorescent lights on (**Figure 1A, B**). Pictures were taken at 1 Hz frequency in order to avoid photodamage. In the primary culture preparation, the presynaptic boutons were patched by looking at the eGFP fluorescence signal (see above). All the axonal varicosities included in the analysis correspond to intact axons; no cut axons (so-called 'axonal blebs') were recorded.

Recordings in slices were done in an upright Olympus BX51W equipped with a ×60, 1.0 numerical aperture objective (NA). Experiments were done at near-physiological temperature (≈34°C) with a Peltier system (Luigs & Neumann). Electrophysiological recordings from primary cultures were done in

an inverted Olympus IX71 microscope equipped with a ×40 objective (NA 0.95). For experiments in slices, epifluorescence excitation was by light-emitting diode (LED) controlled by an OptoLED system (Cairn Research) at 572-/35 nm excitation and at 630/60 nm emission. Filters and dichroic were from Chroma Corporation (Vermont, USA). For experiments in cultures epifluorescence was by an LED light (Light Engine SOLA, Lumencore) at 450-/40 nm excitation and 510-/50 nm emission; fluorescent images were taken with Andor cameras (EMCCD Andor Ixon or a sCMOS Zyla 4.2).

Salts were either from Sigma-Aldrich or Nacalai Tesque (Japan).

## Photolysis

Glutamate was photolysed from MNI-Glutamate (4-methoxy-7-nitroindolinyl-caged-L-glutamate; Tocris Biotechne, UK) with a 405-nm laser (Obis, Coherent, USA) following well-established procedures (*Trigo et al., 2009*). Briefly, MNI-Glutamate was added directly to the bath at a final concentration of 500 µM and photolysed with 100–200 µs and 1–3 mW laser pulses. Duration and power of the laser pulses were adjusted to obtain laser-evoked EPSPs similar to spontaneous ones in terms of amplitude and time course. It is well known that laser power decreases exponentially with depth (*Trigo et al., 2009*). In order to be certain that the same amount of glutamate was released at each tested dendritic varicosity, only dendritic uncaging sites localized at the same imaging plane were chosen (see inset in *Figure 2E*).

## Axonal AP waveform

The voltage waveforms used in the experiments presented in *Figure 5A* correspond to either a single AP (which corresponds to a real AP recorded from an interneuron axonal varicosity) or to the AP preceded by two EPSPs (real EPSPs recorded from an interneuron axonal varicosity and concatenated with the AP in a text file). The width of the AP waveform used, 780 µs, was compatible with the width recorded in the cell-attached configuration (*Figure 1D, E*). The resting $V_m$ of the AP waveform is −70 mV and the peak of the AP is at 40 mV; the EPSPs interval is 19.5 ms, the EPSPs amplitude −58 (#1st EPSP) and −53 mV (2nd EPSP) and the duration of the depolarization (time between the onset of the EPSP and the onset of the AP) is 40 ms. In the experiments where the axonal AP waveform was used (*Figure 5A–C*), the two waveforms (with and without the prior EPSPs) were applied in an interleaved fashion in order to avoid any effect due to the time-dependent washout on the $ICa^{++}_{pre}$.

## Analysis

EPSPs were detected with a threshold detection algorithm implemented by Dr. Taro Ishikawa as IgorPro extensions (TaroTools; https://sites.google.com/site/tarotoolsregister/), and the selection of each event was visually confirmed before subsequent analysis. The EPSPs' CR represents the ratio of the peak axonal and peak somatic depolarizations.

The distance between the center of the soma and the recorded varicosity was measured offline from reconstructions of the recorded cell with Fiji (*Schindelin et al., 2012*). The presynaptic spike widths and amplitudes (*Figures 1, 3, and 4*) in cell-attached recordings were measured from peak to peak, which correspond to the maximal slopes of the rising and decaying phases of the AP voltage waveform: the negative and positive peaks of the presynaptic AP were detected with the same threshold detection algorithm used for the EPSP detection; the time difference between both positions is the CA-spike width; the amplitude difference is the CA-spike amplitude.

## Kinetic simulation of biophysical model for Ca⁺⁺ channels

The normalized steady Ca⁺⁺ current amplitude ($ICa^{++}_{amp}$) at a given membrane potential $V_m$ can be expressed by the following equation,

$$I_{Ca^{++}amp} = P_{Cav(steady\ op)} \cdot (Vm - E_{Ca^{++}}) \tag{1}$$

where $P_{Cav\ (steady\ op)}$ is the steady probability of Ca⁺⁺ channel opening at a certain $V_m$, and $E_{Ca^{++}}$ is the equilibrium potential for Ca⁺⁺.

$P_{Cav\ (steady\ op)}$ was simply expressed using two parameters, $V_{Cav50}$ and $\alpha$, the membrane potential for the half-maximal $Ca_v$ opening and the factor reflecting the voltage sensitivity of channel activation, respectively:

$$P_{\text{Cav}(\text{steady op})} = \frac{1}{1 + e^{-\alpha \cdot (V_m - V_{\text{Cav}50})}} \tag{2}$$

Ca$_v$ channels contain four repeats of structural assembly, each of which has a voltage-sensing S4 domain. In order to study the voltage-dependent state change of each voltage sensor, we simply assumed that the four sensors are independently activated by voltage with an identical probability, $P_{\text{s}(\text{act})}$, and that the channel conducts only when the four voltage sensors are simultaneously active. Thus, the probability of the four sensors being in their active position simultaneously can be represented as the 4th power of $P_{\text{s}(\text{act})}$. From **Equation 2**, the probability of voltage-dependent steady activation of each voltage sensor, $P_{\text{s}(\text{steady act})}$, can be represented as the 0.25th power of $P\text{Cav}_{(\text{stady op})}$, as follows:

$$P_{\text{s}(\text{steady act})} = \left( \frac{1}{1 + e^{-\alpha \cdot (V_m - V_{\text{Cav}50})}} \right)^{0.25} \tag{3}$$

The voltage-dependent state change between active and non-active states of an individual sensor takes place as defined by the velocity ($v$),

$$\frac{dP_{\text{s}(\text{act})}}{dt} = v \cdot \left( P_{\text{s}(\text{steady act})} - P_{\text{s}(\text{act})} \right) \tag{4}$$

where $v$ is assumed to be the product of maximal rate constants for activation or inactivation ($k_{\text{on}}$ or $k_{\text{off}}$, respectively) and the voltage-dependent probability of the sensor state, that is, $v = k_{\text{on}} \cdot P_{\text{s}(\text{steady act})}$ (if $P_{\text{s}(\text{steady act})} > P_{\text{s}(\text{act})}$), or $v = k_{\text{off}} \cdot (1 - P_{\text{s}(\text{steady act})})$ (if $P_{\text{s}(\text{steady act})} < P_{\text{s}(\text{act})}$).

According to the abovementioned assumptions, the (non-conductive) closed state of a single channel can assume four different conformations, 'a' to 'd' (**Figure 6B**, bottom): the four voltage sensors are in their resting or non-activated position (a); there is 1 (b), 2 (c), or 3 (d) of the voltage sensors in their activated position. The voltage-dependent probability of appearance of these individual states of Ca$_v$ ($P$(a), $P$(b), $P$(c), and $P$(d)) were calculated at each time point based on the $P_{\text{s}(\text{act})}$ as follows:

$$P(\text{a}) = (1 - P_{\text{s}(\text{act})})^4 \tag{5}$$

$$P(\text{b}) = 4P_{\text{s}(\text{act})} \cdot (1 - P_{\text{s}(\text{act})})^3 \tag{6}$$

$$P(\text{c}) = 6P_{\text{s}(\text{act})}{}^2 \cdot (1 - P_{\text{s}(\text{act})})^2 \tag{7}$$

$$P(\text{d}) = 4P_{\text{s}(\text{act})}{}^3 \cdot (1 - P_{\text{s}(\text{act})}) \tag{8}$$

The above equations were numerically solved by the Euler method with a time step (d$t$) of 0.01–0.02 ms (as the sampling rate used in the experiments) using the Python programming language. The d$t$ value was set so as to minimize the calculation error. The Python scripts used and the calculated data are available as Supplementary files online.

Based on the assumption that the ECa$^{++}$ is 60 mV, fitting of the Ca$^{++}$ current characterized by the I–V curve (**Figure 6B**) and the ICa$^{++}$ upon a square pulse depolarization pulse (**Figure 6A, C**) yielded $V_{\text{Cav}50} = -17$ (mV), $\alpha = 0.2$, $k_{\text{on}} = 0.55$ (/ms), and $k_{\text{off}} = 0.65$ (/ms), respectively, which were used for the rest of the simulations.

## Statistics

Data are presented as mean ± SD unless otherwise stated. Statistical significance was tested with the Wilcoxon signed-rank test for paired and the Wilcoxon–Mann–Whitney test for non-paired data. The difference between groups was considered significant when $p < 0.05$; when significant, the exact p value is indicated in each figure. In **Figure 4D, E** (axonal CA-spike width and amplitude vs firing frequency), the correlation was assessed by the Spearman rank correlation test. In **Figure 5—figure supplement 2**, statistical significance between the means of the different groups was assessed with the Tukey and Neuman–Keuls post hoc tests.

## Acknowledgements

The authors thank Alain Marty for the critical reading of the manuscript. Funding: Agence Nationale pour la Recherche (ANR); JCJC grant ANR-17-CE16-0011-01 (FFT); International Brain Research Organization (IBRO); Return Home Fellowship (FFT); Japan Society for Promotion of Science, Core-to-Core Program A (SK); Japan Society for Promotion of Science, KAKENHI grants 22H02721 and 22K19360 (SK); Takeda Science Foundation (SK); and Naito Foundation (SK).

## Additional information

### Funding

| Funder | Grant reference number | Author |
| --- | --- | --- |
| Agence Nationale de la Recherche | JCJC grant ANR-17-CE16-0011-01 | Federico Trigo |
| International Brain Research Organization | Return Home Fellowship | Federico Trigo |
| Japan Society for the Promotion of Science | Core-to-Core Program A | Shin-ya Kawaguchi |
| Japan Society for the Promotion of Science | KAKENHI grants 22H02721 | Shin-ya Kawaguchi |
| Japan Society for the Promotion of Science | KAKENHI grants 22K19360 | Shin-ya Kawaguchi |
| Takeda Science Foundation | | Shin-ya Kawaguchi |
| Naito Foundation | | Shin-ya Kawaguchi |

The funders had no role in study design, data collection, and interpretation, or the decision to submit the work for publication.

### Author contributions

Federico Trigo, Conceptualization, Data curation, Formal analysis, Supervision, Funding acquisition, Validation, Investigation, Visualization, Methodology, Writing – original draft, Project administration; Shin-ya Kawaguchi, Conceptualization, Data curation, Software, Formal analysis, Supervision, Funding acquisition, Investigation, Methodology, Project administration, Writing - review and editing

### Author ORCIDs

Federico Trigo (ID) http://orcid.org/0000-0001-6704-7303
Shin-ya Kawaguchi (ID) http://orcid.org/0000-0002-8386-1185

### Ethics

This study was performed in strict accordance with the recommendations in the Guide for the Care and Use of Laboratory Animals of the National Institutes of Health, USA. All procedures were approved by the local committee for animal experiments in Université de Paris (approval number 750607), in IIBCE (approval number 001-01-2023), and in Graduate School of Science, Kyoto University (approval number 202213).

### Decision letter and Author response

Decision letter https://doi.org/10.7554/eLife.85971.sa1
Author response https://doi.org/10.7554/eLife.85971.sa2

## Additional files

### Supplementary files
• MDAR checklist
• Source code 1. Python code for the simulation of Cav sensor activations presented in *Figure 6*.

## Data availability

All the data supporting the findings of this study presented in figures are available in *Figure 1—source data 1*, *Figure 2—source data 1*, *Figure 3—source data 1*, *Figure 4—source data 1*, *Figure 5—source data 1*, and *Figure 6—source data 1*. The Python code for simulation of Cav sensor activations is uploaded as Source code 1 and also is available at the authors website: http://www.nb.biophys.kyoto-u.ac.jp/model/Python_CaVSim.zip.

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
