## [Editor Report]

This paper shows compelling experimental evidence for a novel mechanism for analog transmission of synaptic information from dendrites to distal nerve terminals in CNS interneurons. Brief subthreshold depolarizations caused by small EPSPs in dendrites are capable of activating the voltage sensors of presynaptic Ca2+ channels in nerve terminals, thus leaving them in a transient "primed" state. A subsequent suprathreshold action potential can then open these Ca2+ channels more efficiently leading to transmitter release facilitation. The effect decays within a few milliseconds and it is not dependent on residual Ca2+ levels or changes in presynaptic action potential waveform.

---

## [Decision Letter]

**Decision letter after peer review:**

Thank you for submitting your article "Analogue signaling of somato-dendritic synaptic activity to axon enhances GABA release in young cerebellar molecular layer interneurons" for consideration by *eLife*. Your article has been reviewed by 3 peer reviewers, and the evaluation has been overseen by a Reviewing Editor and Gary Westbrook as the Senior Editor. The reviewers have opted to remain anonymous. The reviewers have discussed their reviews with one another, and the Reviewing Editor has drafted this to help you prepare a revised submission.

Essential revisions:

No new experiments are required but more analysis and modeling is recommended to support the conclusions of the paper. Please address the following two items:

1) Provide more detail on Rs values and the quality of the voltage clamp of Ca currents. For example, provide a plot of Ca current peak amplitudes (or current charge) against Rs to determine if these are correlated or not. Summarize this analysis in the methods or as a new supplementary figure.

2) Further modeling (as shown in Figure 6) should be attempted to demonstrate the plausibility of the Ca channel gating scheme as an explanation for the facilitation of the release. Provide further evidence that the short depolarizations can indeed drive the several closed states of the Ca channel to an open state. Otherwise, the conclusions need to be toned down and perhaps Figure 6 should be removed if the authors cannot provide it with further modeling support.

*Reviewer #1 (Recommendations for the authors):*

Based on the outstanding technical quality, the convincing conclusions, and the novel and innovative proposed mechanism, we strongly support publication of the manuscript. However, we have the following comments:

Time-dependent modeling of calcium channel gating should be provided to demonstrate that the proposed mechanism can indeed explain the experimental data. We would suggest to analyze the gating of calcium channels during APs (1.) with pre-APs, (2.) without pre-APs, and (3.) with pre-APs with a 3-ms-gap.

*Reviewer #2 (Recommendations for the authors):*

The following points may be considered when revising the manuscript.

1) The time resolution of the traces is not so fast in Figure 2. It will be good to show sub-threshold responses in a ms time scale, so that one can see the delay and attenuation by eye.

2) It is a bit pity that most critical experiments (looking at Ca currents, the last figures) have been done in culture. However, it is technically demanding and clamp is difficult in slices. The authors should explain more explicitly why these sets of experiments are done in culture and others in slices.

3) The authors cannot exclude the two possibilities, perhaps. (a) Ca dependent facilitation of Ca channels. They can be relatively short-lived, and sensitive only to very concentration of Ca buffers (expression of high endogenous Ca buffers). (b) Some sort of voltage clamp problem. The authors assume perfect clamp at the terminal. Maybe activation, deactivation kinetics of Ca currents, and the clamp speed (electrical properties) can be reanalyzed from the existing data.

*Reviewer #3 (Recommendations for the authors):*

1) I don't think the term "analog-digital coupling" is very self-explanatory. Please consider finding a better description for example "analog and AP coding of axons" or "analog coding via subthreshold somatic potentials" etc.

2) The author frequently use the qualifiers "axon" or "axonal" or even "axonal presynaptic terminal" when in fact referring to presynaptic compartments, presynaptic ICa(V) or presynaptic AP waveform. The authors never directly recorded from axons which likely have very different properties than boutons/terminals. Please change your terminology for clarification, i.e. use 'presynaptic ICa(V)' and 'presynaptic AP' etc.

3) When recording in whole-cell configuration from presynaptic boutons, the authors use high resistance pipettes (18 or 25 MOhm) which results in very high access resistance (83 MOhm on average). This is to some degree unavoidable when recording from such tiny structures as MLI varicosities but it nevertheless impacts the recordings. Therefore, the whole issue deserves some more detailed description and discussion. The authors correctly emphasize that because of the 10fold smaller capacitance of varicosities compared to somata, adequate voltage clamp speed may still be achieved, especially because the recorded whole-terminal currents are generally quite small. What was the upper threshold in terms of Rs for accepting recordings? Was there any series compensation applied during voltage clamp and if so how much and what was the resulting uncompensated Rs? Current-clamp recordings are also expected to be heavily low-pass filtered unless 100% bridge compensation was used with the HEKA amplifiers. Perhaps such filtering can partially explain the large scatter of AP widths in Figure 1D?

4) During the experiments shown in Figure 1C, both soma and boutons were held in voltage-clamp. But likely only the soma was depolarized. If so, please move the command-voltage waveform on top of the somatic recording and call the INa(V) recorded in the bouton an escaping sodium current as it escaped the presynaptic voltage clamp at Vhold.

5) Please supply some more motivation for performing the Glu uncaging experiments illustrated in Figure 2E,F. What additional information is gained by the uncaging that could not be gained by simply injecting depolarizing current waveforms in to somata of recorded cells or by recordings such as those illustrated in Figure 2A-D. At present, the reader is a bit left alone with figuring out why Glu uncaging near dendrites was performed for studying axonal signaling of subthreshold somatic potentials to boutons.

6) In Figure 4 F-H, no second AP was fired in response to two somatic depolarizations in very close succession (<2.2 ms in the example in panel F) presumably due to refractoriness. For ease of understanding, that refractory period should also show up in panels G and H as a blank space (i.e. missing numbers) and the time axis should start at 0 ms inter-stimulus interval.

7) In Figure 5B, an average increase in presynaptic ICa(V) following small depolarizations of ~55% was calculated from 4 cells showing very little increase (~30%) and two other cells showing a large increase >100%. If the ICa(V) facilitation is due to an effect on channel gating, as the authors suggest, one would expect a very reproducible phenomenon in voltage-clamp recordings. This does not seem to be the case. Why? Different subtypes of presynaptic VGCCs? Different classes of MLI boutons?

8) In Figure 6C (y-axis scale bar is missing), the authors measure variance in order to support their argument that VGCCs do not open during the small conditioning depolarization. The mean current does not reveal any increase. Why do the authors think that current variance would reveal channel opening which does however not show up in the mean current trace?

9) In Figure 6E,F, the authors present model fits for the ICa(V) I-V curve and presumed state occupancies of a kinetic scheme depicted in panel D. It is not clear to me how the authors produced these traces. Are these simulations using a kinetic scheme such as that presented in Li, Bischofberger and Jonas (2007) J.Neurosci.. If so, the model parameter should be presented. If not, the authors should implement such kinetic model, fit it to their data and demonstrate that such scheme is able to reproduce the ICa(V) facilitation in response to small conditioning prepulses which they observe experimentally. That would strongly strengthen the authors' conclusion about the proposed mechanism of IC(V) facilitation. Without such kinetic simulations, it remains speculation.

10) In the paragraph about "Ideas and speculation" that authors seem to suggest that the phenomenon they studied is vanishing with maturation. Even if that is the case, it would be more comforting for a reader to be given a 'more positive outlook'. The authors further suggest that strong somato-axonal coupling may play a role during maturation of GABAergic contacts but fail to indicate how that could work.

11) Finally, I strongly advice referencing and discussion additional previous work, especially work on presynaptic calyx terminals which seems strangely completely absent from the manuscript. Strong candidates include (but are not limited to) for example:

(i) regarding propagation of subthreshold depolarizations to terminals

Paradiso and Wu (2008) Nature Neurosci. "Small voltage changes at nerve terminals travel up axons to affect action potential initiation" (should really be referenced)

(i) regarding presynaptic ICa(V)

Borst and Sakmann (1998) J.Physiol. "Calcium current during a single action potential in …"

Borst and Sakmann (1998) J.Physiol. "Facilitation of presynaptic calcium currents in …"

Li, Bischofberger and Jonas (2007) J.Neurosci. "Differential Gating and Recruitment of P/Q-, N-, and R-Type ca^2+^ Channels in …"

Lin, Oleskevich and Taschenberger (2011) "Presynaptic ca^2+^ inﬂux and vesicle exocytosis at the mouse endbulb of Held …"

(ii) regarding presynaptic AP waveform and its relationship to release probability

Sabatini and Regehr (1997) J.Neurosci. "Control of Neurotransmitter Release by Presynaptic Waveform at …"

Borst and Sakmann (1999) PhilTransRSocLondB "Effect of changes in action potential shape on calcium currents and transmitter release in …"

Taschenberger and von Gersdorff (2000) J.Neurosci. "Fine-Tuning an Auditory Synapse for Speed and Fidelity …"

Li, Bischofberger and Jonas (2007) J.Neurosci. "Differential Gating and Recruitment of P/Q-, N-, and R-Type ca^2+^ Channels in …"

Boudkkazi, Fronzaroli-Molinieres and Debanne (2011) J.Physiol. "Presynaptic action potential waveform determines …"

---

## [Author Response]

Essential revisions:No new experiments are required but more analysis and modeling is recommended to support the conclusions of the paper. Please address the following two items:1) Provide more detail on Rs values and the quality of the voltage clamp of Ca currents. For example, provide a plot of Ca current peak amplitudes (or current charge) against Rs to determine if these are correlated or not. Summarize this analysis in the methods or as a new supplementary figure.

We have performed further analysis on the relationship between Rs and Ca currents, as suggested. Furthermore, we have extended the discussion and we have given more explanation in the Materials and Method section. Finally, we have incorporated a new Supplementary Figure (Figure 6—figure supplement 1) where we show that there is no correlation between the kinetics of the ICa^++^ pre and the series resistance values.

2) Further modeling (as shown in Figure 6) should be attempted to demonstrate the plausibility of the Ca channel gating scheme as an explanation for the facilitation of the release. Provide further evidence that the short depolarizations can indeed drive the several closed states of the Ca channel to an open state. Otherwise, the conclusions need to be toned down and perhaps Figure 6 should be removed if the authors cannot provide it with further modeling support.

We thank reviewers for pointing out an important simulation analysis. As suggested, we have performed time-dependent simulation of Ca channel activation upon an AP or a large square pulse depolarization, with or without coupling with various patterns of pre-pulses. The results are now summarized in a new Figure 6. Our simulation clearly supports our idea that brief subthreshold depolarization partially activate some of Cav voltage sensors, priming the opening of Cav upon the following suprathreshold depolarization. Importantly, such an effect disappears very quickly when an interval of several ms is given between the pre-pulse and the AP (or large depolarization pulse).

Reviewer #1 (Recommendations for the authors):Based on the outstanding technical quality, the convincing conclusions, and the novel and innovative proposed mechanism, we strongly support publication of the manuscript. However, we have the following comments:

We appreciate the very supportive positive comments by the reviewer. We have responded to all the helpful comments provided by three reviewers, as described below.

Time-dependent modeling of calcium channel gating should be provided to demonstrate that the proposed mechanism can indeed explain the experimental data. We would suggest to analyze the gating of calcium channels during APs (1) with pre-APs, (2) without pre-APs, and (3) with pre-APs with a 3-ms-gap.

According to the constructive suggestion, we have performed simulation of Ca channel activation with time-dependent change upon an AP with or without pre-pulses, and the results are now shown in a new Figure 6. Our results support the idea that the subthreshold depolarization right before the AP tends to ‘prime’ the channels by bringing some of the voltage sensors to the activated state. This priming rapidly disappears in several milliseconds.

Reviewer #2 (Recommendations for the authors):The following points may be considered when revising the manuscript.1) The time resolution of the traces is not so fast in Figure 2. It will be good to show sub-threshold responses in a ms time scale, so that one can see the delay and attenuation by eye.

Following the reviewer’s recommendation, we have added a new panel in Figure 2D showing examples of individual, simultaneous somatic and axonal EPSPs in a ms time scale.

2) It is a bit pity that most critical experiments (looking at Ca currents, the last figures) have been done in culture. However, it is technically demanding and clamp is difficult in slices. The authors should explain more explicitly why these sets of experiments are done in culture and others in slices.

As it is mentioned in the text, the Ca current data shown in Figure 6A and B does include recordings from acute slices. Data were similar in culture and slice, so they are pooled together in the analysis. In the text, we have added more explanation to specifically mention this situation (p17, line 425). Concerning the results presented in Figure 5, we tried hard to obtain paired recordings from a presynaptic varicosity and its postsynaptic partner in the slice. Unfortunately, however, we never succeeded in getting a stable basket cell bouton>Purkinje cell recording and thus turned to the primary culture model. We added a short sentence to specifically mention this (p14, lines 356 to 362).

3) The authors cannot exclude the two possibilities, perhaps. (a) Ca dependent facilitation of Ca channels. They can be relatively short-lived, and sensitive only to very concentration of Ca buffers (expression of high endogenous Ca buffers). (b) Some sort of voltage clamp problem. The authors assume perfect clamp at the terminal. Maybe activation, deactivation kinetics of Ca currents, and the clamp speed (electrical properties) can be reanalyzed from the existing data.

Thank you for pointing out our insufficient explanation. Regarding the possibility (a) of, that is, Ca-dependent Ca channel facilitation, here the subthreshold pre-pulse did not produce any Ca influx, but the Ca current upon the immediately following AP was nevertheless augmented. Thus, the predominant mechanism working here is different from Ca-dependent Ca current facilitation. Rather, our results may imply that the Ca current facilitation and the resultant synaptic facilitation observed during the repetitive high-frequency presynaptic AP firings might be partly mediated by the Ca channel ‘priming’ mechanism in addition to the Ca-mediated Ca current facilitation. We thank the reviewer for giving us the insightful comment to think about that possibility. As for the second possibility (b), the reviewer’s concern is reasonable. Therefore, we have added explanation about the potential concern arising from the incomplete voltage-clamp condition in the text (p24-25). Concerning the additional simulation, we have carefully analyzed the Ca current activation and de-activation to fit the biophysical model of Ca channels, and the model simulation nicely captures the essential experimental results, mainly the ‘priming’ effect of sub-threshold depolarization on Ca channels. Thus, now we believe that our data, with support of the new simulation results, demonstrate the mechanism of the analogue signaling involves a modulation of Ca channel states that affect release upon an AP arriving at the presynaptic terminal.

Reviewer #3 (Recommendations for the authors):1) I don't think the term "analog-digital coupling" is very self-explanatory. Please consider finding a better description for example "analog and AP coding of axons" or "analog coding via subthreshold somatic potentials" etc.

We are sorry that our explanation about the term “analog-digital coupling” was not sufficient in the previous manuscript. However, the term is widely used to unify the different mechanisms that have been described in the literature to highlight that the subthreshold somatodendritic activity transmitted to the axon can affect AP-dependent release. To make it clearer, following the reviewer’s comments, we have nevertheless added explanations in Introduction (p3, lines 61 to 64) and Discussion (p22) to clarify this point.

2) The author frequently use the qualifiers "axon" or "axonal" or even "axonal presynaptic terminal" when in fact referring to presynaptic compartments, presynaptic ICa(V) or presynaptic AP waveform. The authors never directly recorded from axons which likely have very different properties than boutons/terminals. Please change your terminology for clarification, i.e. use 'presynaptic ICa(V)' and 'presynaptic AP' etc.

The reviewer is right that we never recorded from the axon proper, sometimes also called main axonal trunk, but from presynaptic axonal boutons; both structures have certainly different properties, both anatomically and physiologically. We have changed the terminology all along the text in order to follow the reviewer’s suggestion.

3) When recording in whole-cell configuration from presynaptic boutons, the authors use high resistance pipettes (18 or 25 MOhm) which results in very high access resistance (83 MOhm on average). This is to some degree unavoidable when recording from such tiny structures as MLI varicosities but it nevertheless impacts the recordings. Therefore, the whole issue deserves some more detailed description and discussion. The authors correctly emphasize that because of the 10fold smaller capacitance of varicosities compared to somata, adequate voltage clamp speed may still be achieved, especially because the recorded whole-terminal currents are generally quite small. What was the upper threshold in terms of Rs for accepting recordings? Was there any series compensation applied during voltage clamp and if so how much and what was the resulting uncompensated Rs? Current-clamp recordings are also expected to be heavily low-pass filtered unless 100% bridge compensation was used with the HEKA amplifiers. Perhaps such filtering can partially explain the large scatter of AP widths in Figure 1D?

The quantification of the presynaptic AP width has been done exclusively from recordings in the cell-attached configuration (Figure 1D) to avoid the errors associated with the pipette capacitance in small structures (Ritzau-Jost et al., Cell Rep). The large scatter of presynaptic AP widths in the young age group has already been described by Rowan et al. and Begun et al. (with electrophysiology and voltage dye imaging) and is probably due to a lack of maturation of voltage-dependent conductances at this age range. With maturation the variability goes down, which is also an indication that the large scatter in the presynaptic AP width it is not an artifactual finding. However, we agree with the reviewer that the voltage-clamp speed and the associated space-clamp deserves more explanation. We have added a paragraph in the Discussion section (p24 and 25). The average, uncompensated series resistance value was 82 Mohm, with a SD of 31 Mohm. Rs values were compensated by 30%. The recordings from the boutons usually lasted less than 5 to 6 minutes, and the Rs values remained constant during these time. However, only the experiments in which the Rs values did not change more than 20% were used. These points have also been explained in the Methods (p32, lines 759 to 768).

4) During the experiments shown in Figure 1C, both soma and boutons were held in voltage-clamp. But likely only the soma was depolarized. If so, please move the command-voltage waveform on top of the somatic recording and call the INa(V) recorded in the bouton an escaping sodium current as it escaped the presynaptic voltage clamp at Vhold.

We have changed the sentence (p4) and figure, as suggested.

5) Please supply some more motivation for performing the Glu uncaging experiments illustrated in Figure 2E,F. What additional information is gained by the uncaging that could not be gained by simply injecting depolarizing current waveforms in to somata of recorded cells or by recordings such as those illustrated in Figure 2A-D. At present, the reader is a bit left alone with figuring out why Glu uncaging near dendrites was performed for studying axonal signaling of subthreshold somatic potentials to boutons.

Based on the results published in 2012 by Abrahamsson et al. (Neuron), who showed that in more mature MLIs the dendritic filtering of synaptic potentials is not negligible, we decided to perform glutamate uncaging because this is the only way to activate physiologically the dendritic postsynaptic receptors in situ and therefore to accurately quantify the distance-dependent reduction of dendritic EPSPs while they propagate down the dendrites and the axon. The results presented here in the younger age group indicate that the amount of dendritic filtering at this age range is negligible; however, this was not known beforehand. In addition, our data exhibited, more importantly, that such different sizes of EPSPs similarly reached the axonal presynaptic boutons. Following the reviewer’s suggestion, we have added to the manuscript sentences explaining the reasons to perform Glu uncaging experiments (p6, lines 161 to 166).

6) In Figure 4 F-H, no second AP was fired in response to two somatic depolarizations in very close succession (<2.2 ms in the example in panel F) presumably due to refractoriness. For ease of understanding, that refractory period should also show up in panels G and H as a blank space (i.e. missing numbers) and the time axis should start at 0 ms inter-stimulus interval.

We have changed the time scale of Figure 4G and H, according to the reviewer’s suggestion.

7) In Figure 5B, an average increase in presynaptic ICa(V) following small depolarizations of ~55% was calculated from 4 cells showing very little increase (~30%) and two other cells showing a large increase >100%. If the ICa(V) facilitation is due to an effect on channel gating, as the authors suggest, one would expect a very reproducible phenomenon in voltage-clamp recordings. This does not seem to be the case. Why? Different subtypes of presynaptic VGCCs? Different classes of MLI boutons?

As the reviewer correctly points out, the ICa++ and PSCs (Figure 5C) measured in 2 of the cells show a dramatic increase when comparing the test (with a pre-pulse depolarization before AP) vs control conditions, while in 4 of the cells the increase is more subtle. Similar diversity was also observed at the pre-pulse-caused PSC augmentation in slice (see Figure 3C). We do not know the reason for such a heterogeneity and we can only speculate. As our new simulation results indicate, the slow kinetics of Ca channels is very critical for the ICa augmentation (Figure 6G). Thus, even slight difference of the Ca channel modulation and/or subtypes may largely affect how the augmentation operates. As for the diverse effects, we have added explanation in the text (p25, first paragraph).

Previous work indicates that the amount of vesicles in the RRP is limited (Trigo et al., PNAS 2012; Pulido et al. Neuron 2015) and that the amount of postsynaptic receptor saturation is high (Auger and Marty Neuron 1997: Auger et al. J. Neuroscience 1998). These 2 aspects set an upper limit to the increase in the PSC amplitude. However, we have recorded synapses where postsynaptic receptor saturation does not seem to be present (mainly axo-somatic synapses, ie: between a presynaptic MLI bouton and a postsynaptic soma, or synapses where the postsynaptic neuron is a Purkinje cell). We tend to think that this may partly explain the heterogeneity observed in Figure 5C. Concerning the VGCC, their exact nature has only been studied in MLI presynaptic terminals with 2P calcium imaging, but not with electrophysiology. According to these studies, different types of VGCC contribute to release from these terminals (N and P/Q type and more recently, also L-type [Rey et al., J. of Neurochemistry, 2020)] but their exact role under different paradigms of stimulation and their distribution are not known.

8) In Figure 6C (y-axis scale bar is missing), the authors measure variance in order to support their argument that VGCCs do not open during the small conditioning depolarization. The mean current does not reveal any increase. Why do the authors think that current variance would reveal channel opening which does however not show up in the mean current trace?

As pointed out by the Reviewing editor (item 2) and the reviewers (reviewer1 Major comment, reviewer 3 point #9), this work lacks direct experimental evidence for the priming of the Ca channels. Our work proposes a new mechanism by which subthreshold depolarizations reaching the presynaptic terminals can affect release in MLIs. To test this idea, according to the reviewers’ recommendation, we have performed additional simulation, and observed such a ‘priming’ effect of Ca channels by sub-threshold depolarization (Figure 6). Thus, we have omitted the variance data which was shown in the previous version of manuscript, because it was used to indirectly support the idea of some ‘priming’ phenomenon just by emphasizing the lack of direct channel activation.

9) In Figure 6E,F, the authors present model fits for the ICa(V) I-V curve and presumed state occupancies of a kinetic scheme depicted in panel D. It is not clear to me how the authors produced these traces. Are these simulations using a kinetic scheme such as that presented in Li, Bischofberger and Jonas (2007) J.Neurosci.. If so, the model parameter should be presented. If not, the authors should implement such kinetic model, fit it to their data and demonstrate that such scheme is able to reproduce the ICa(V) facilitation in response to small conditioning prepulses which they observe experimentally. That would strongly strengthen the authors' conclusion about the proposed mechanism of IC(V) facilitation. Without such kinetic simulations, it remains speculation.

According to the constructive suggestion, we have performed simulation of Ca channel activation, and the results are now shown in new Figure 6. Our results support the idea that subthreshold depolarization tends to ‘prime’ the channels by putting some of voltage sensors in an activated state, which is rapidly gone in several milliseconds.

10) In the paragraph about "Ideas and speculation" that authors seem to suggest that the phenomenon they studied is vanishing with maturation. Even if that is the case, it would be more comforting for a reader to be given a 'more positive outlook'. The authors further suggest that strong somato-axonal coupling may play a role during maturation of GABAergic contacts but fail to indicate how that could work.

We have done experiments in the mature age group that show that the EPSP coupling is significantly reduced and we are currently performing experiments to understand the mechanism (and/or the physiological relevance) of such a reduction. Nevertheless, we have followed the reviewer’s advice and have rewritten that section (p25-26) in order to give a “more positive outlook” in the “ideas and speculations” section, by highlighting the potential role in developmental maturation of GABAergic synapses.

11) Finally, I strongly advice referencing and discussion additional previous work, especially work on presynaptic calyx terminals which seems strangely completely absent from the manuscript. Strong candidates include (but are not limited to) for example:

We thank the reviewer for these suggestions and apologize for not making a thorough citation of appropriate literature. We have made an effort to cite important articles throughout the text and the discussion.